# Occurrence and Characterization of Paraffin Wax Formed in Developing Wells and Pipelines

**Marwa M. El-Dalatony [1], Byong-Hun Jeon [1,*], El-Sayed Salama [2], Mohamed Eraky [3], Won Beom Kim [4], Jihoon Wang [5] and Taewoong Ahn [6,*]**

[1] Department of Earth Resources and Environmental Engineering, Hanyang University, Seoul 04763, Korea; marwa@hanyang.ac.kr

[2] Department of Occupational and Environmental Health, School of Public Health, Lanzhou University, Lanzhou 730000, Gansu, China; salama@lzu.edu.cn

[3] Environmental Biotechnology Department, City of Scientific Research and Technology Applications, New Borg El Arab City, Alexandria 21934, Egypt; memo_science86@hotmail.com

[4] Research institute, Golden Engineering Co. Ltd., Seoul 05836, Korea; goldeneng.kim@gmail.com

[5] Department of Petroleum and Natural Gas Engineering, New Mexico Tech, Socorro, NM 87801, USA; jihoon.wang@nmt.edu

[6] Petroleum and Marine Research Division, Korea Institute of Geoscience and Mineral Resources (KIGAM), Daejeon 34132, Korea

* Correspondence: bhjeon@hanyang.ac.kr (B.-H.J.); twahn@kigam.re.kr (T.A.); Tel.: +82-2-2220-2242 (B.-H.J.); +82-4-2868-3307 (T.A.)

**Abstract:** Deposition and precipitation of paraffin wax in pipelines are major problems in the production, transfer, storage, and processing of crude oil. To prevent complete clogging, it is necessary to minimize and remove deposited wax in pipelines and tubing. Significant research has been done addressing the mechanisms of wax formation and its composition. In this review, the status of research and perspectives on the occurrence and characterization of the paraffin wax that forms in crude oil developing wells and pipelines has been critically reviewed. Several approaches for detecting paraffin wax and managing wax formation damage during oil recovery were discussed. This review also highlighted the effects of temperature and crude oil type on wax formation.

**Keywords:** waxy crude oil; micro- and macro-crystalline paraffin; wax appearance temperature (WAT); formation damage; saturates, aromatics, resins, and asphaltenes (SARA); rheological properties

## 1. Introduction

Crude oil is the major energy source all over the world. It contains thousands of different hydrocarbons that are classified into four fractions: saturates (aliphatics), aromatics, resins and asphaltenes. The properties of crude oil vary with temperature, where some of petroleum fluids cause wax crystals precipitation at low temperature [1]. The wax precipitates form deposits on pipe walls, causing difficulties in extraction, as well as problems in maintaining a pipeline's capacity for crude oil transport [2]. Crude oil contains significant amount of wax in the range of 3–44% which would crystallize and precipitate during extraction and transportation, and cause higher oil viscosity and higher pour point values. Elevation of these values results in pressure drop, gelation, lower flowability, and higher pumping cost [3,4]. Waxes are mostly aliphatic and nonpolar compounds which have a high molecular weight, limited solubility in crude oil, and can aggregate or associate in solution [5,6]. The wax deposited in pipelines and tubing may create costly separation problems. It is necessary to understand the mechanism and factors that contribute to the formation and stabilization of such emulsions for both economic and environmental development [7]. Global petroleum corporations are facing huge challenges in wax removal because of the decrease in cavitation bubbles within the

production tubing, which lead to high cost of maintenance and low productivity of the 'wells' [1]. Recently, research has been conducted to address the mechanisms of wax formation and its composition in pipelines [8–10]. In this review, the current status of research activities and perspectives on the occurrence and characterization of paraffin wax formed in developing wells and pipelines were well-reviewed. Several approaches for detecting wax and for reducing the rate of its deposition in pipelines were highlighted. The factors affecting wax formation were also discussed.

*1.1. Wax Deposition*

Wax deposition refers to the formation and eventual growth of solid layers on the surface which is attached to the crude oil. Wax deposits can be formed from dissolved wax molecules through molecular diffusion [11]. Wax deposition in the pipe can only take place when the temperature of inner pipe wall is below the wax appearance temperature (WAT). The deposited molecules of wax which precipitated near the pipe wall begin to develop gel at the cold surface. The formed gel is a 3-D wax crystals mass and contains a substantial amount of oil immersed within it [12]. During normal conditions of cooling, the inner pipe wall has a lower temperature compared to the bulk oil. Therefore, the precipitation degree of waxy compounds is generally greater on the inner wall than in the bulk, however dissolved waxy components are greater in the bulk oil than on the inner pipe wall. Such phenomena creates a gradient in radial concentration of the waxy components between the bulk oil and the wall [13]. The concentration gradient causes a diffusion of waxy compounds from the bulk oil towards the wall [11].

Several studies have been conducted to understand the interactions among composition of reservoir rocks (such as carbonate rock or/and sandstone), tubing wall materials (metallic, ceramic or micanite), composition of crude-oil (salts, organic acids, alcohols, and other surface activating reagent), as well as types of produced wax (macro-crystalline wax and micro-crystalline wax). Two major factors have been reported to affect the wettability: (1) rocks surface morphology and (2) the intermolecular surface forces among the three phases (i.e., oil, rock, and brine). Surface wettability has a great effect on the formation of severe wax layers. The wettability of any system refers to the relative interaction between fluids (oil/water) and the solid phase (rock surface). The hydrophobic forces of a surface are reduced when the wettability is reduced, which promote the initiation of wax deposition. The rock reservoirs were commonly categorized as water-wet, oil-wet, or intermediate-wet based on the rock affinity toward oil or water phase [14]. Such interactions are generally categorized into two classes: (a) non-polar (Lifshitz-van der Waals interactions) and (b) polar (acid-base interactions). Arsalan et al. [14] established a new method to measure these major interactions through characterization of the surface energetics of carbonate rocks and some sandstone by inverse gas chromatography.

*1.2. Mathematical Modelling of Wax Deposition*

There are four wax deposition mechanisms that need to be taken into account for modeling the wax deposit growth in pipelines, i.e., molecular diffusion, shear dispersion, gravity settlement and Brownian diffusion [15,16]. The molecular diffusion, which can be described by taking the mass transfer and energy balance into account, is considered as the dominant mechanism during the wax deposition process [15,17–20]. Burger, Perkins and Striegler [15] and Weingarten and Euchner [21] proposed that the shear dispersion mechanism needs to be incorporated for the wax deposition behavior analysis, especially in laminar flow. However, Brown, Niesen and Erickson [18] performed a set of experiments and pointed out that the mechanism does not take a role during the wax deposition. The gravity settling mechanism suggests that the wax crystals are would be precipitated to the bottom of the pipelines as the crystals are denser than the oil. However, it appears that the effect is negligible as experimental studies comparing vertical and horizontal fluid flows did not reveal any difference in the deposited wax amount [22]. As soon as the wax crystals are precipitated and suspended in the oil, the wax crystals will behave with Brownian motion. Since the motion effect is likely to transport the crystals to the area with lower wax concentration, it may have an influence on the wax deposition behavior. However, its effect is generally considered minimal and it is generally neglected during

modeling the overall deposition mechanisms [22]. Consequently, the latter three mechanisms are widely not accepted, especially for the computational modeling procedures [22–28]. In this part of the study, the most dominant mechanism, the molecular diffusion, is briefly reviewed.

In order to incorporate the molecular dispersion mechanisms, the radial temperature gradient of the pipeline needs to be taken into account. This is because a radial convective flux of wax molecules is induced by the concentration gradients, which strongly depends on the temperature gradient inside the pipeline. Therefore, the flux of the oil and the wax deposits can be measured by calculating the difference in the concentration of wax between the bulk and the interface. The mass transfer can be expressed with the following equation:

$$\frac{d}{dt}\left(\pi\left(R^2 - r_i^2\right)\overline{F_w}(t)L\rho_{gel}\right) = 2\pi r_i L k_m (C_{wb} - C_{ws}) \tag{1}$$

where, $R$ is the clean pipe radius, $r_i$ is the flowable radius, $\overline{F_w}(t)$ is weight fraction of solid in the wax deposit, $L$ is the pipeline length, $\rho_{gel}$ is the wax deposit density, $k_m$ is the convective mass transfer coefficient, $C_{wb}$ and $C_{ws}$ are the dissolved paraffin concentration in the bulk and at the oil-gel interface, respectively.

Another process that needs to be incorporated is the growth of the deposited wax. Since there exists a temperature gradient within the deposit, an internal mass transfer also occurs. The wax deposition growth can be described by the following Equation:

$$-2\pi r_i \overline{F_w}(t)\rho_{gel}\frac{dr_i}{dt} = 2\pi r_i k_m (C_{wb} - C_{ws}) - 2\pi r_i \left(-D_e \frac{dC_w}{dr}\bigg|_i\right) \tag{2}$$

where, $D_e$ is effective diffusivity of wax in the waxy given by Cussler, et al. [29]:

$$D_e = \frac{D_{wo}}{1 + \frac{\alpha^2 \overline{F_w}^2}{1 - \overline{F_w}}} \tag{3}$$

where, $D_{wo}$ is the molecular diffusivity of wax in oil and $\alpha$ is the aspect ratio of the wax crystals in the deposit.

Within the system the energy balance is met as the heat conduction across the deposit thickness is the sum of the latent heat of solidification and the radial convective heat flux. Therefore, the energy balance can be expressed with the following equation:

$$2\pi r_i h_i (T_b - T_i) = \frac{2\pi k_e (T_i - T_a)}{ln(R/r_i)} - 2\pi r_i k_l (C_{wb} - C_{ws}(T_i))\Delta H_f \tag{4}$$

where, $T_b$ is the temperature of oil at the center, $T_i$ is the oil-wax interface temperature, $T_a$ is the ambient temperature, $k_e$ is the conductive heat coefficient, $h_i$ is the inner convective heat transfer coefficient and $\Delta H_f$ is the heat of wax solidification.

Assuming flow in the pipe is laminar, the heat transfer coefficients can be determined by the empirical correlations, such as the Seider and Tate correlation [30], and the Hausen correlation [31]. The former is valid for long tubes, while the latter shows a good agreement with a short tube length [20]. The Nusselt number for the Graetz number lower than 100 can be calculated by the Hausen correlation as follows:

$$Nu_i = 3.66 + 1.7813 \cdot 10^{-3} \cdot \left(\frac{(Gz_i)^{\frac{5}{3}}}{\left(1 + 0.04 \cdot (Gz_i)^{\frac{2}{3}}\right)^2}\right), \; i = h \; or \; m \tag{5}$$

where, $Gz_h$ and $Gz_m$ are Graetz number for heat transfer and for mass transfer, respectively. When the Graetz number is larger than 100, the Nusselt number can be obtained by the Seider and Tate correlation as follows :

$$Nu_i = 1.24 \cdot (Gz_i)^{\frac{1}{3}}, \ i = h \ or \ m \tag{6}$$

The heat transfer coefficient and the mass transfer coefficient can be then calculated by the following equations:

$$h_h = \frac{Nu_h \cdot k_{oil}}{2R} \tag{7}$$

$$k_m = \frac{Nu_m \cdot D_{wo}}{2R} \tag{8}$$

where, $k_{oil}$ is the thermal conductivity of the oil.

The computational modeling can be a powerful tool to predict and to interpret the wax deposition phenomena after validated with available experiments data. However, most of the lab-scale experiments are based on the single-phase flow, which frequently yields deviated outcomes when compared to more-general multiphase flow cases. In addition, Leporini et al. [25] emphasized that the modeling results are not always applicable for the field scale analysis and that the scale effect also needs to be taken into account for more reliable interpretation. Consequently, recent studies focus on more complex environments, such as wax deposition behavior under multi-phase flow and turbulent flow.

### 1.3. Composition and Characterization of Waxy Crude Oil

The wax in petroleum crudes exists in various phase states (gas, liquid, or solid) depending on pressure and temperature [32]. It consists of paraffin hydrocarbons (known as paraffin wax) and naphthenic hydrocarbons (known as *iso*-paraffin wax) (Table 1). Paraffin wax has a density of ~900 kg/m$^3$ and its a heat capacity is of ~2.14–2.9 J·g$^{-1}$·K$^{-1}$, and it is also recognized as a macro-crystalline waxes [33]. On the other hand, iso-paraffin wax, which is produced during the crude oil refinery process, is identified as a micro-crystalline wax. Iso-paraffin wax has better engine-combustion characteristics due to its high melting point and molecular weight [34]. The crystal structure of micro-crystalline waxes is smaller and thinner than that of macro-crystalline waxes, making them more flexible. Macro- and micro-crystalline waxes have different functional properties (including viscosity and melting point) due to the differences in their hydrocarbon content (linear or branched). Oliveira et al. [35] investigated the influence of micro- and macro-crystalline paraffin on the properties of crude oil and found that Mic-wax can be dissolved or dispersed in solvents and display a sharp transition of gel-strength. The micro-crystalline wax usually crystallizes at a higher temperature compare to macro-components where stronger gels are formed. This phenomenon might be due to the fact that micro-components possess lower number of carbons compare to macro-components (Figure 1).

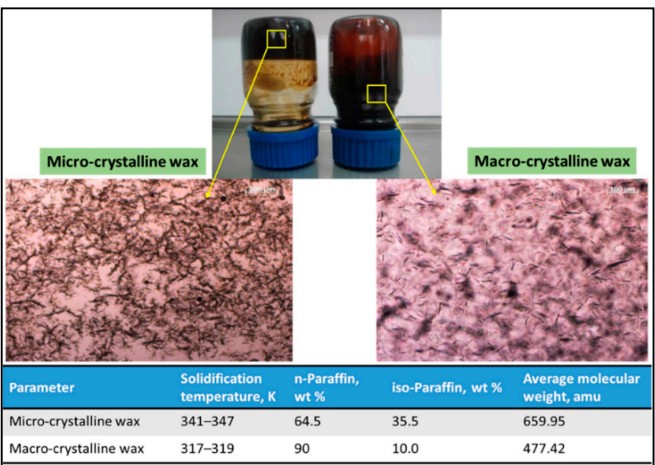

| Parameter | Solidification temperature, K | n-Paraffin, wt % | iso-Paraffin, wt % | Average molecular weight, amu |
|---|---|---|---|---|
| Micro-crystalline wax | 341–347 | 64.5 | 35.5 | 659.95 |
| Macro-crystalline wax | 317–319 | 90 | 10.0 | 477.42 |

**Figure 1.** The impact of micro- and macro-crystalline paraffin on the characteristics of crude oil and organic solvents. Adapted with permission from [35], Lviv Polytechnic, 2005.

**Table 1.** Composition of petroleum (hydrocarbons and non-hydrocarbons) adapted from [4,36].

| Hydrogen Family | Most Important Hydrocarbons | Chemical Characteristics | Comments |
|---|---|---|---|
| **Hydrocarbons** | | | |
| Naphthenes | Cyclopentane, methyl cyclopentane, dimethylcyclopentane cyclohexane, and 1,2 dimethylcyclohexane | 5–6 carbon atoms in the ring | ➢ $R_n$ is the number of naphthenic rings. <br> ➢ Normal crude oil contains approximately 50% naphthenes by weight. |
| Paraffins **(Alkanes)** | Methane, ethane, propane, butane, pentane, and hexane | Straight carbon chain | ➢ The boiling point rises as the number of carbon atoms rises. <br> ➢ Paraffin turn into wax, as carbon numbers ranges 25–40. |
| Iso-paraffins **(Iso-alkanes)** | Isobutane, isopentane, neopentane, isooctane | Branched carbon chain | ➢ The number of isomers rises with increasing the carbon atoms number. |
| Olefins **(Alkenes)** | Ethylene | One pair of carbon atoms | ➢ Olefins do not exist in crude oil; however, they are formed during oil processing. <br> ➢ Olefins are undesirable in the end product due to high reactivity. <br> ➢ Olefins with low molecular weight have better antiknock characteristics. |
| Aromatics | Benzene, toluene, xylene, ethyl benzene, cumene, and naphthalene | Six carbon atoms in a ring, with three around the linkage | ➢ Aromatics are not necessary in kerosene and lubricating oil. <br> ➢ Because benzene is carcinogenic, it is an undesirable component of gasoline. |
| **Non-hydrocarbons** | | | |
| Oxygen compounds | Naphthenic acids and Phenols | NM | ➢ The content of these compounds is 2%. These acids not only cause corrosion problems at various stages of processing, but also cause pollution problems. |
| Sulphur compounds | Hydrogen sulphide and mercaptans | NM | ➢ These are undesirable due to their foul odor (0.5% to 7%). |
| Nitrogen compounds | Quinoline, pyridine, pyrrole, indole, and carbazole | NM | ➢ Upon exposure to sunlight, nitrogenous compounds in kerosene and gasoline degrade the color of the product. <br> ➢ The effect of causing gum formation is normally less than 0.2%. |

NM = Not mentioned.

The characteristics of crude oil plays a significant role in its production, transport, refining, and storage. Hence, it is necessary to understand and predict the performance of waxy crude oil and their wax to avoid various problems of pipelines plugging and clogging. Several observations from industrial field have shown frequency of wax formation during production of the oil crude [1]. Countries such as Kuwait, Qatar, Uganda, and South Sudan have waxy crude oil in their reserves [37]. A drastic decrease in productivity from 30,000 Barrels of Oil Per Day (bopd) to zero within 24 h was observed in the North Sea Miller field [38]. The Alba field located on the UK continental shelf and the Zakum field located in Abu-Dhabi, are included among the world's largest oil fields where sulphate and carbonate scale were found [39,40]. More information about the composition and characterization of waxy crude oils in different regions is shown in Table 2. Most of the countries exhibit a good relationship between wax content, API gravity, wax appearance temperature (WAT), and pour point.

WAT is the temperature at which wax crystal formation begins. The formation of wax causes the fluid to become cloudy, hence WAT is also called cloud point. The industry established techniques for measuring WAT in the laboratory ranged from ASTM D2500, cross polar microscopy (CPM), light transmission (LT), differential scanning calorimetry (DSC), viscometry, and filter plugging infrared (FTIR) to cold finger tests [41,42]. The resulting values of WAT depend on the measurement technique used. The ASTM method relies on the optical observation of wax crystals, which reduces the reliability of WAT measurements of opaque or dark oils. Therefore, CPM, LT, DSC, or viscometry methods are preferred for WAT measurements of dark oils. DSC and viscometry are affected by the amount of precipitated wax, while light transmittance (LT) and light scattering are affected by the number of wax crystals. WAT measurement methods have advantages and disadvantages, so it is recommended to use multiple technologies together depending on the given conditions and circumstances [41,43]. WAT is not only related directly with crude oil wax content, but also with its paraffin distribution [44]. Although waxy crude oil shows challenging effects during its production, it yields valuable end products, as it can be refined and transformed into gasoline, fuel oils, kerosene, petroleum naphtha, diesel fuel, jet fuel, heating oil, liquefied petroleum gas, and asphalt base [45]. Propylene and ethylene can also be generated directly by cracking crude oil without the use of naphtha.

**Table 2.** Wax contents, API gravity, WAT, and pour point values of waxy crude oil from diverse regions.

| Region | Wax Content (wt, %) | API * Gravity | WAT (°C) | Pour Point (°C) | Reference |
|---|---|---|---|---|---|
| China | 18.25 | 24.2 | - | 43 | [46] |
| Dulang, Malaysia | 3 | 12.6 | 31 | 33.76 | [47] |
| Angsi, Malaysia | 2 | 42.6 | 28 | 33.32 | |
| South America | NM ** | 27 | 36.4 | 9 | [48] |
| Eastern Egyptian | 3.3–4.5 | - | - | - | [49] |
| Upper Egypt | 11.92 | 31.6 | - | 27 | [50] |
| South East Asia | 18–38 | 25–40 | 26–68 | 15–60 | [51] |
| North Sea Crude oil | 15 | 33 | 42 | 27 | [52] |
| Venezuelan (Boscan) | 4.1 | - | - | - | [53] |
| Russian | 9.4–12.2 | - | - | - | |
| Sudan | 21.2 | - | - | - | [54] |
| Gulf of Mexico | 7.8 | | | | [55] |
| Mexico (PC) | 11.26 | 36 | - | −30 | [56] |
| Mexico (IRI) | 10.91 | 28.4 | - | −26 | |
| Iran | 13.1 | 34.9 | - | 26 | [57] |
| India | 22.4 | 44.2 | - | 22 | [58] |
| China (Changqing) | 20.78 | 34 | - | 30 | [59] |

* API = American Petroleum Institute; ** NM = Not mentioned.

### 1.4. Saturates, Aromatics, Resins, and Asphaltenes Distribution and Density of Crudes

Most of the world's oil reserves contain waxy crude oil, which is characterized as either sweet or sour, depending on its sulfur content. Sour waxy crude oil is more expensive to refine than sweet waxy crude oil, because of its higher sulfur content. However, based on crude oil's physicochemical properties (e.g., polarity and boiling-point temperature), it can be subdivided into 4 major fractions: saturates, aromatics, resins, and asphaltenes (SARA) [60]. Asphaltenes, which have the highest molecular weight, have a significant role in the deposition of organics during petroleum processing and production [61]. There are two mechanisms of asphaltene precipitation concerning the nature of asphaltenes in solutions (Figure 2a).

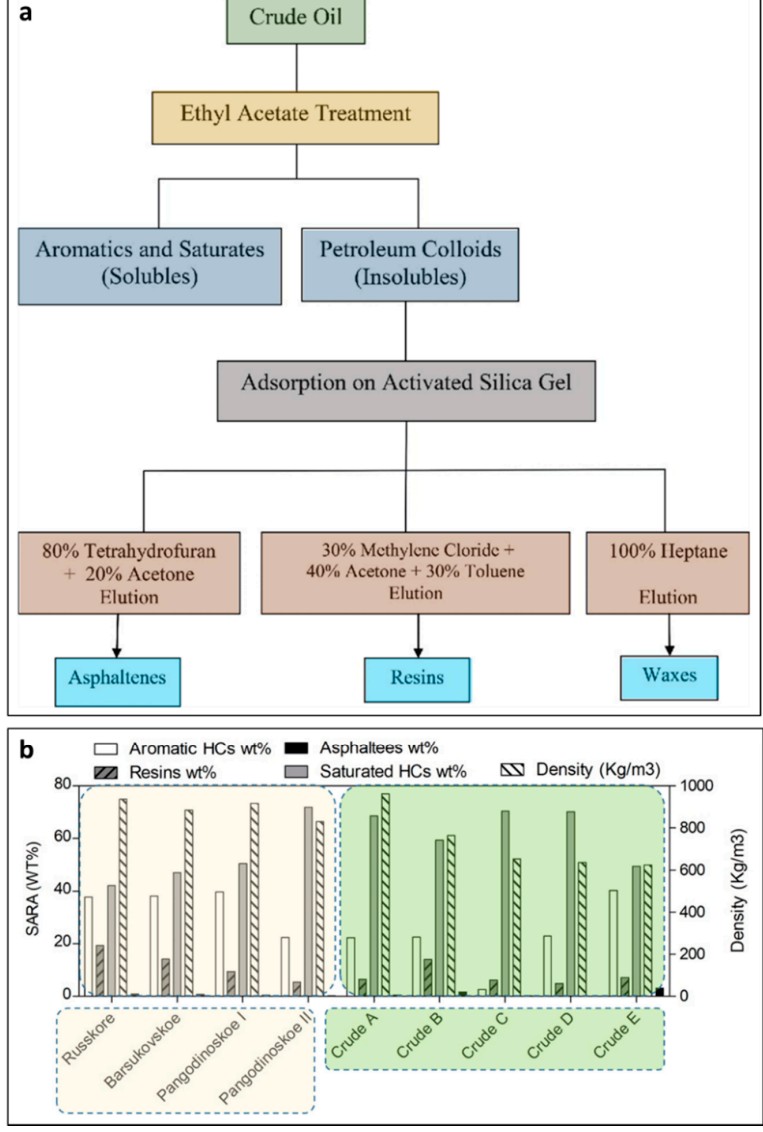

**Figure 2.** The mechanisms of asphaltene precipitation concerning the nature in solution (**a**) and the distribution of saturated hydrocarbons (HCs), aromatics, resins, and asphaltenes and the density of some crude oils from different regions (**b**). The data highlighted in yellow refer to the Middle East [62], while those in green are for Northern West Siberia [63].

The first mechanism proposed asphaltenes to be in a liquid state and dissolved in oil, while in the other one, asphaltenes are solid particles suspended in colloids crude oil [64]. Some asphaltenes fractions including toluene and benzene are soluble. However, n-pentane and n-heptane are

insoluble [65]. Asphaltenes have the highest percentage of heteroatoms (with O, S, and N) and organometallic constituents (with Ni, V, and Fe) and can be suspended as micro-colloid particles. Each particle comprises one or more layers of asphaltene monomers, with attached resin (as surfactants to stabilize the colloid in suspension) [66]. The hydrogen-to-carbon ratio in wax is ca. 2, while that in asphaltenes is ca. 1.153 [33].

The standard analytical acetone method [67] is an industrial practice to measure the wax content of crude oils. However, to evaluate the content of wax at various temperatures, several runs are required to be performed. The spectroscopic methods used for wax detection are less time-consuming and less complicated than the high performance liquid chromatography (HPLC) method [66]. Resin molecules stabilize the asphaltene molecules, inhibiting their major aggregation in petroleum. Aromatics surround resins and warrant a progressive move to the petroleum bulk where saturated hydrocarbons are the major content [68]. The saturate, aromatic, and resin content of a crude oil are determined simply using infrared and near-infrared spectroscopy at high pressure and temperature [66]. The composition and density of saturated hydrocarbons and aromatic ones, and the structural-group properties of resins and asphaltenes in four heavy, highly viscous oils of Northern West Siberia [63] and five light crude oils with low asphaltene content from the Middle East region [62] are shown in Figure 2b.

## 2. Problems Caused by Waxy Crude Oil

### 2.1. Formation Damage during Oil Recovery

In the field of petroleum engineering, the formation damage normally indicates formation permeability deterioration that causes additional pressure drop during fluid transportation through porous media. The term "formation damage" is principally used to reflect the damage that occurs from well activities during crude oil processes such as drilling, cementing pipe, perforating, and production. The damage occurs from either movement and bridging of fine solids or chemical reactions that result in precipitates and changes in wettability [69]. The most common type of formation damage from waxy crude oil is organic deposition, which is composed mainly of paraffin and contains some asphaltenes, or resin. Inorganic materials such as sand, clay, and corrosion residues are also found in the organic deposition [70]. Mixed scales (i.e., inorganic and organic) occur simultaneously in the same system. Inorganic scales caused by calcium, barium, and strontium salts ($CaCO_3$, $CaSO_4$, $BaSO_4$, and $SrSO_4$) sometimes appear as a mixture with paraffin wax scale [71]. Mixed scales may result in highly complex structured deposits that are difficult to treat and thus require aggressive, severe and sometimes costly remediation techniques [72]. Such scales are formed due to the decrease in tubing pressure, release of $CO_2$, and evaporation of water.

Formation damage occurs with inadequate well operations such as incompatible fluid chemistry, hot oiling, or injections of cold fluid. Paraffin deposition can occur in reservoir rocks, which can cause severe formation damage and consequently decrease reservoir productivity [73]. The main causes of paraffin deposition are environmental changes (including temperature, pressure, and loss of dissolved gases) affecting solution equilibrium [74] and field operations (such as hot oiling, water flooding, and cold fluid injection) that are used to improve oil recovery (Table 3) [75]. As production progresses, the oil shows a pressure drop, which might lead the light components of the oil to evaporate. Thus, the local temperature would decrease in reservoir rocks near the wellbore. Paraffin crystals flocculate and accumulate as the oil flows through the rocks, plugging pores and causing formation damage [76,77]. Alaska North Slope (ANS) is one of the major oil and gas reserves located in an Arctic environment in the United States, accounting for approximately 15% of the nation's oil production [15]. The continued production of ANS oil faced challenges, including wax deposition due to the decline in production rate. Wax deposition in ANS showed associated technical issues such as permeability reduction, formation damage, reduction in the interior diameter, changes in the reservoir fluid composition, increased pressure drop, and limiting the operating capacity of the entire

production system. UK Lasmo abandoned and withdrawn its platform in November 1994 due to repeated wax blockage. The US Minerals Management Service published 51 severe wax-related reports between 1992 and 2002 in Gulf of Mexico flow lines. The main concern raised upon the wax deposition was the enhanced capital investment and operating costs which caused serious financial strain on the operator due to blockage of facilities by wax deposits. Reduced production and the additional cost of controlling and managing wax placed a greater risk of abandonment on such fields [78].

Fluid injection into the reservoir is a common production technique for oil recovery improvement and production stimulation (Table 3). For acidizing, fracturing, and water flooding, fluids are injected at lower temperatures than the reservoir temperature and can cool the area around the well to below the cloud point [79]. Consequently, paraffin crystals start to precipitate and deposit on the pore space, impeding the fluid flow through the rock pores. Hot oiling can cause wax deposition due to the invasion of contaminated fluid. Oils used in hot oiling include paraffin or alkane components that cause paraffin precipitation at a certain pressure and temperature condition due to their high molecular weight. The injected hot oil dissolves paraffin components, which increases the wax content of the oil. Subsequent cooling of the initial hot oil causes successive wax deposition. Acidizing, $CO_2$ flooding, and natural gas liquid (NGL) flooding can induce organic deposition including wax by injection of incompatible fluids such as acid, $CO_2$, hydrocarbon components, etc. [80,81]. Injection of chemicals along with water will make new chemical deposit, which can decrease the flow between layers, induce volume of water injected, improve the degree of recovery, and enhance the production efficiency [82]. Nevertheless, this type of chemical reaction will produce pollution in oil and the capacity for water interference would be damaged. Polymer flooding is one of the major oil recovery methods to recover oil remained from the conventional retrieval processes which increase the fluid viscosity, improve the efficiency of volumetric sweep, and thereby further enhance the oil recovery factor. However, the use of polymer flooding for long time will effect on the reservoirs through accumulation of scales which is one of the major challenges in the development of oil and gas fields. Gas could dislocate oil by either miscible or immiscible displacement where some major driving mechanisms occurred (such as viscosity reduction, reservoir swelling, and interfacial tension reduction) [82]. Carbon dioxide is the most commonly used gas for miscible displacement as it decreases the viscosity of oil and it is cheaper than liquefied petroleum gas. The miscible process, is valid to light to medium gravity crude oils, however the immiscible method, can be applied to heavy gravity oils. A miscible displacement process maintains the pressure for the reservoir and improves oil displacement [83]. $CO_2$ is mainly effective in deep reservoirs (>2000 ft.), where it is in a supercritical state. Miscible and immiscible $CO_2$ flooding in a fractured oil field have been evaluated in a reservoir modeling approach, where the average field pressure and oil recovery factor were found to be $1.041 \times 10^8$ stb and 5095 psia, respectively [84]. This variety of technology is recommended in order to enlarge the reservoir volume and improve the efficiency of recovery, but unless natural $CO_2$ exists in the neighborhood area, it's commonly hard to collect sufficient amounts of $CO_2$ for industry use.

### 2.2. Mechanisms of Formation Damage by Paraffin Wax

There are two mechanisms by which wax precipitation can cause formation damage: (1) deposition that plugs or attaches to pore walls and (2) high viscosity of crude oil caused by cohesion [81]. The paraffin deposition begins when the smallest paraffin crystals precipitate. Crude oil flows through rock pore channels, which allow the small crystals to be close to the pore wall. With sufficient wettability of the pore wall, these small crystals will settle and accumulate, thus reducing the pore throat size and leading to formation damage [85,86]. As mentioned earlier, paraffin crystallization can also cause formation damage by increasing the viscosity of the crude oil, even if it is not deposited on the pore walls. In addition, paraffin crystals tend to form gels. Gel formation enhances both the cohesion and adhesion of crystals [87] and may result in a sudden increase in the viscosity of the crude oil. Paraffin crystallization and deposition elevate the crude oil viscosity and decrease the formation permeability, correspondingly changing the crude oil mobility in the reservoir formation.

**Table 3.** Causes of formation damage by paraffin wax [88–91].

| | Cold Fluid Injection | Contaminated Fluid Invasion | Cooling by Gas Expansion | High Flow Rate Through Formation |
|---|---|---|---|---|
| | Acidizing work | | High The gas/oil ratio well | Flowing well |
| Operation | Fracturing work | | $CO_2$ flooding | Steam flooding |
| | Water flooding | Hot oiling work | Natural gas liquid (NGL) flooding | |

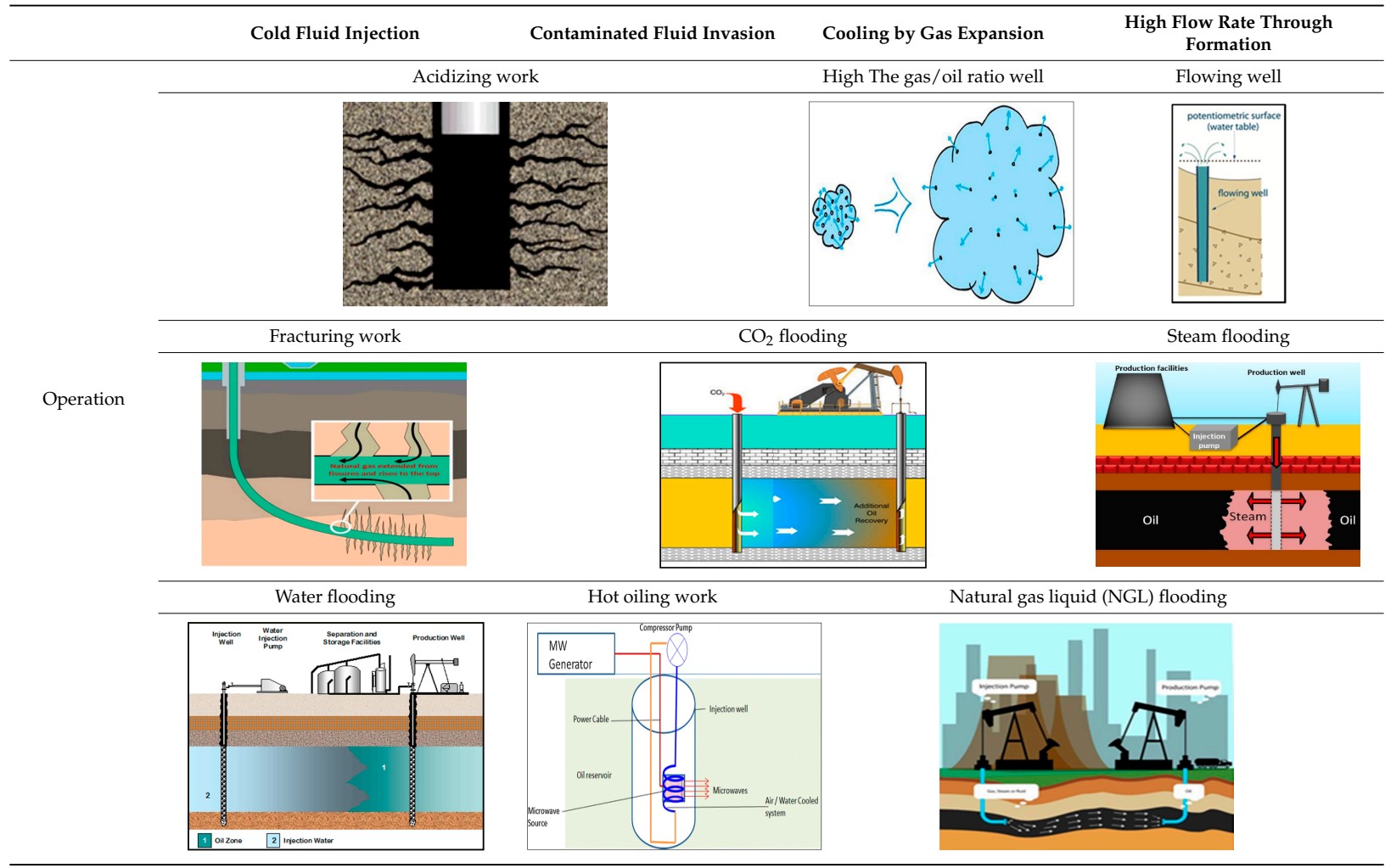

The mobility in a porous media is generally expressed as:

$$m = \frac{k}{\mu} \tag{9}$$

where $m$ is the fluid mobility (md/cp), while k and $\mu$ are the formation permeability (md) and the fluid viscosity (cp), respectively [92]. According to Equation (9), both increase in crude oil viscosity and decrease in formation permeability prompt a reduction of crude oil mobility. Ultimately, this phenomena would induce additional pressure drops, resulting in unfavorable hydrocarbon recovery.

### 2.3. Effects of Wax Deposition on the Flow Assurance of Hydrocarbon

Hydrocarbons plays a major role in the global energy market where a feasible transport of oil and gas is secured. Flow assurance is defined as the process of ensuring the successful and economical flow of hydrocarbon stream transported from the reservoir to the point of sale. Maintaining the hydrocarbons flow during oil and gas production is a critical task, especially in deep-water or cold weather, due to long distances, high pressures and low temperatures [25,93]. Flow assurance is applied to the petroleum flow route during every stages of production. Waxes, hydrates, asphaltenes, naphthenates, scales, corrosion, erosion, and emulsions are the major flow assurance issues that need to be considered [12]. Wax deposition is the primary flow assurance challenge, which create pressure abnormalities and causes an artificial blockage. Therefore, understanding wax and wax related process is required for the development of new solutions for field developments in artic and remote locations, and also for deeper and colder fields [93]. Various methods for prevention and/or removal of paraffin wax deposits have been performed to deal with wax problems. These methods include active heating, fused-chemical reactions, insulation, pigging, chemical treatments, wax-repellent surfaces cold flow technologies, and biological treatments [13].

Wax deposition is effected by temperature and critical carbon number (CCN) [94]. CCN is the number of carbon for heavy paraffin which is soluble in the liquid phase of the initial deposit. The CCN has been categorized into upper CCN and lower CCN. Analysis of a deposit in the field approves the aging performance of paraffin deposits, and shows that values of CCN are in the mid 20's at field conditions [95]. Paraffin constituents which have a number of carbon atoms higher than the CCN, diffuse from the bulk oil to the gel deposit of liquid phase, consequently depositing and increasing the solid wax content of the deposit. However, paraffin constituents which contain a carbon atoms less than the CCN, diffuse out of the deposit. A value of CCN can be measured by comparing the content of an aged gel deposit and its corresponding bulk fluid, using a device such as a spinning disk or a cold finger [96]. The wax deposition rate is rapid initially, however, it slows with increasing the amount of wax deposited on the pipe surface. The thickness of the wax layer (macro-crystallization) gets bigger and acts as thermal insulation factor. The wax deposition and its thickness control by several mechanism such as Shear dispersion, Brownian diffusion, gravity settling, and molecular diffusion [97].

Wax precipitation leads to flow restriction during the oil flow, which consequently causes problems when trying to restart the flow [98–101]. If oil production is halted by an emergency situation or a scheduled maintenance, the solubility and temperature of wax decreases in a static condition. In this circumstances, wax-oil gel is formed due to precipitation of wax molecules, which can lead to blockage of the entire pipeline. In order to restart the oil flow and ensure a stable oil transport, the gel must be broken down. This waxy oil pipeline restart problem is challenging especially at temperature below the pour point at which the liquid oil remains pourable. When solving this problem, it is important to evaluate the pressure to break the plug of wax-oil gel. In order to predict the breakdown pressure, it is essential to estimate the gel strength [101]. Therefore, it is necessary to understand the formation and breakdown phenomena of wax-oil gel in order to provide a reliable flow assurance solution for gelled pipeline restart.

### 2.4. Paraffin Wax Management

The approaches for managing paraffin crystallization and deposition in oil fields can be broadly categorized into: (1) mechanical, (2) thermal, (3) chemical, and (4) microbial [102]. The mechanical method is used to remove paraffin that has been deposited inside the pore walls of tubing and other equipment (such as flow lines, pipeline, and separators) mechanically using devices known as pigs, scrapers or cutters [103]. The efficacy of the pigging process can vary extensively, depending on pigs design, as well as other parameters [73,104]. A technique of high pressure water spray was proposed for wax removal using less supply of water. The use of high pressure water sprays as mechanical method, followed by injection a mixture of sterling beads to enhance erosion has been applied [105]. Previous efforts have been provided to Al Ghawar of Saudi Aramco, the world's largest oil field for descaling iron sulphide ($Fe_7S_8$) formed in one of the gas fields, using the sterling beads [1]. Recent technique of using aeration-cavitation in descaling production tubes covered with wax have been implemented. Aeration up to 12% were found to eliminate the cavitation nearly to zero [1]. The thermal method treats the deposited paraffin by increasing the temperature beyond the cloud point. However, hot oil can cause additional paraffin deposition, and it consumes a large amount of energy due to the requirement of injecting hot oil to supply heat [106].

The chemical method is the preferred paraffin control [107]. The chemicals used for this method include dispersants, solvents, crystal modifiers, and surfactants. Solvents are used to dissolve paraffin deposits. Dispersants breakdown paraffin deposits into much smaller particles. Surfactants solubilize the paraffin in oil. Crystal modifiers prevent paraffin crystals from depositing by altering their growth. Crystal modifiers can also play a role in changing rheological properties such as the viscosity and pour point of crude oil [108]. These chemical methods are difficult to standardize, because the paraffin contained in the crude oil has varying chemical composition. Therefore, the chemical method should be specifically designed and used for a particular paraffin or oil, considering chemical compatibility with target formation and the reservoir conditions. As well, the environmental consequence of used chemicals need to be considered to ensure environmental protection where water-based chemicals or environmentally friendly chemicals are used to avoid any potential hazards to humans or to the environment [109]. Microbial methods have been used as an alternative to the conventional paraffin treatment methods as they are non-combustible, non-carcinogenic, non-pathogenic, and environmentally safe [110]. This method uses microorganisms to produce by-products that act as surfactants or solvents to paraffin molecules and dissolve or eliminate deposited paraffin in the formation [111]. However, microbial growth and activity can be limited by reservoir conditions such as temperature, pressure, salinity, and permeability.

The surfactant molecule carries two functional groups, (1) a hydrophilic (polar) or water-soluble and (2) a hydrophobic (non-polar) or oil-soluble. The hydrophobic group is a long chain hydrocarbon (C8–C18), while the hydrophilic group is formed by moieties such as alcohols, sulfates, sulfonates (anionic), carboxylates, quaternary ammonium salts (cationic), and polyoxyethylenated chains (nonionic) [112]. The conventional surfactant is also known as wax dispersant. Prevention of wax deposition was achieved in New Mexico field using wax dispersant which works as inhibitor for wax crystals growth or inhibitor for waxes deposition [60]. When crude oil gets in contact with water or brine, the natural surfactants deposit at the interface and form an adsorbed film that decreases the tension interface of the crude oil/water. Depending on the type of crude oil, the adsorbed film can be either fluid or viscoelastic [113]. Therefore, surface elasticity, surface viscosity, surface charge of the adsorbed film, and the molecular packing are very significant parameters that control coalescence of emulsion droplets and oil drop movement in porous media. Ceramic or micanite in-line ring heaters were recommended to be applied for prevention of wax accumulation in tubing [114].

## 3. Detection of Deposited Wax

### 3.1. Approaches to Determine the Presence of Wax in Crude Oil

Many global crude oils fields contain significant amounts of wax (3~44%) [4]. Determining the wax content of crude oils has great significance for oil production, which is challenging due to poor solubility of high molecular weight paraffin in the usual solvents. One of the most effective method is high temperature gas chromatography (HTGC). The distinction between general GC and HTGC is not clearly defined, but GC at 340 degrees or more can be called HTGC. HTGC enables the analysis of a wide range of molecular weights from C7 to >C100 [115]. Recently, two-dimensional GC with flame ionization detection (HT-GCxGC-FID) or a time-of-flight mass spectrometer (HT-GCxGC-TOFMS) has been used to separate *n*-paraffins, isoparaffins, and cycloparaffins [116]. Widely used and well-reported methods for investigating crude oils characteristics include the standard acetone method in units of production (UOP method) [67] and the differential scanning calorimetry (DSC) method [54]. However, numerous approaches for wax detection have been recently studied, including: (1) pigging and take-out, (2) distillation and extraction, (3) pressure drop and heat transfer, (4) ultrasound and laser spectroscope, and (5) thin-layer chromatography. In pigging and take-out, which is the traditional experimental method for measuring the degree of wax deposits, a part of pipe is removed, and the volume of deposited wax is measured [117]. Distillation and extraction are European standard techniques used for wax content determination in bitumen. Distillation determines the paraffin bitumen wax from a distillate produced in a process at very high temperatures (>500 °C) [53]. The extraction method includes three phases: (1) asphaltene extraction, (2) aromatic compound extraction, and (3) wax crystallization in an ether/ethanol mixture at −20 °C [118]. Methods utilizing pressure drop and heat transfer can be used to measure wax deposits ultimately without down time [30]. Piroozian et al. [119] investigated the patterns of two-phase flows (oil-water flow) on a laboratory scale (at 28 °C). Visual observation showed that, although the system temperature was maintained above the WAT (24 °C) of the dehydrated crude oil throughout the experiment, there was still wax deposition in some conditions. This was due to the effect of emulsified water on the WAT.

The use of ultrasound and laser spectroscope has proven to be successful in determining the presence of small solid particles and paraffin within paraffin-containing oil samples [120]. However, these approaches were only performed on a laboratory scale. The investigation and production strategies in deep sea zones is economically feasible for deep water drilling. Development of oil well industry (~160 miles away from the shore) explores more extensive wax problems due to the enlarged transportation lines on cold ocean level. Practical methods still need to be designed for application of these techniques to existent subsea pipelines. Thin-layer chromatography with flame ionization detection (TLC-FID) has been utilized to describe and detect petroleum-associated resources (such as bitumen and crude oils) [121]. Lu et al. [122] presented a simple technique for detecting wax content in petroleum products. The method depends on TLC-FID and comprises a two-step development with *n*-heptane and methyl ethyl ketone (MEK) as solvents. Saturates are separated from other components (more polar) based on their high solubility in *n*-heptane and low interaction with silica (adsorbent). Then waxes are separated from the saturate portion using MEK at a low temperature, where waxes are present in solid state (Figure 3).

An alternative method of measuring and evaluating wax deposition in pipelines was employed, which determined light absorption through a source of light and a detector circuit attached within a pipe [120]. On a laboratory scale, this detector circuit proved its capability to detect the impurity, even at a very small percentage.

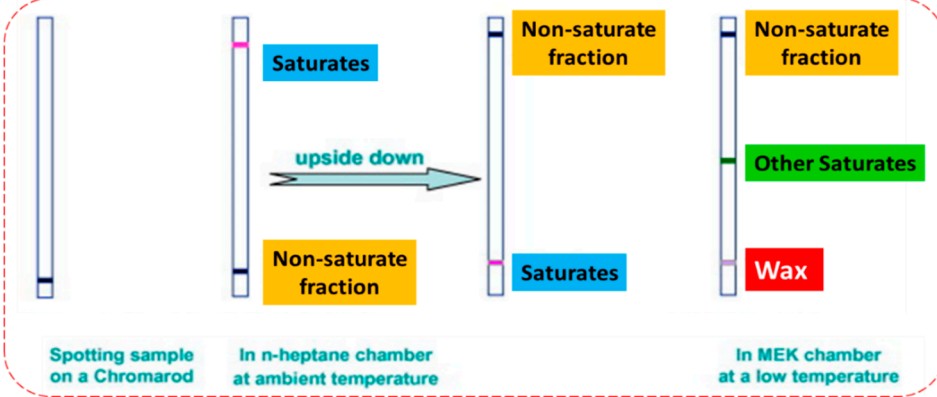

**Figure 3.** Waxes separated from the saturate portion using methyl ethyl ketone (MEK) as a poor solvent at a low temperature at which wax is in solid state.

### 3.2. Detection of Blockage

Total and partial blockage of petroleum production due to wax deposition in subsea pipelines is a main challenge for the industry and is considered to be one of the major risks for deep water production systems [123]. This challenging issue causes significant capital losses and an associated loss of production, along with the necessity to replace the plugged lines [124]. Improving of existing techniques is required for determining the degree of wax deposition and plugs at different points in a pipeline. Pressure echo methods have been used to locate the blockage by evaluating the period taken by a pressure wave to be reflected back in the pipeline from the blockage point [124]. Another way is to pressurize the pipeline to measure the outside diameter of the pipeline using a caliper and video camera. A significant difference in diameter can be noted when upstream pressure is applied in the pipeline is [125]. A blockage map could be created to indicate and characterize the size of blockages based on the length and diameter of the restrictions [126].

### 4. Factors Affecting the Formation of Waxy Crude Oil

### 4.1. Temperature

Temperature plays an important role in the production and transport of waxy crude oil. N-paraffins (C12–C35), components of wax that have an affinity to the walls of pipes, are deposited. Wax molecules behave as liquids under reservoir temperature and pressure, since they are dissolved in the crude oil [94]. The reduction in flowable cross-sectional area due to wax deposition causes additional pressure drops, which further increase the cost of transport. Wax deposition is the most operational and long-standing concern in the petroleum industry and the transport of crude oil through sub-sea pipelines. It is essential to quantify and investigate the factors causing problems in maintaining optimum crude oil productivity and flow assurance, including wax deposition. Elements that affect wax deposition include temperature gradient across the pipeline wall, flow rate, and fluid residence time [96].

The temperature at which paraffin crystal formation begins is the wax appearance temperature (WAT). The WAT of various origin crude oils was in the range of 14.2–37.8 °C depending on the wax content and API gravity (Table 2). Moreover, the crude oils have wide range of physical property of precipitation (−3.42 to 6.39 mW) and melting energy (−2.5 to 7.46 mW) [127]. However, wax formation and deposition on the pipelines wall begin below the WAT [128–130]. The variance between the cold temperature at sub-sea level and the warmer temperature of the internal pipeline fluid is the main cause of wax formation [96]. WAT is considered as the freezing point where the fluids begins to solidify [131], which causes a decrease in pipeline flow as precipitated wax becomes heavier and thicker. Exposure of pipelines to cold surroundings causes accumulation, gathering, and solidification of droplets, giving a cloudy appearance [127]. Therefore, the flow oil ultimately reduce, causing

blockage and high pressure in pipelines. At an ocean floor temperature (4 °C), wax molecules start to deposit on the pipeline walls. However, the heat loss is inconsistent, as there is a difference in temperature gradient within the pipeline, which affects the wax deposition rate [20]. In waxy crude oil, precipitation, deposition, and gelation are primarily responsible for wax deposition. Clogging, due to continuous deposition of waxes, ultimately results in an abrupt shut down of pipelines (Figure 4a).

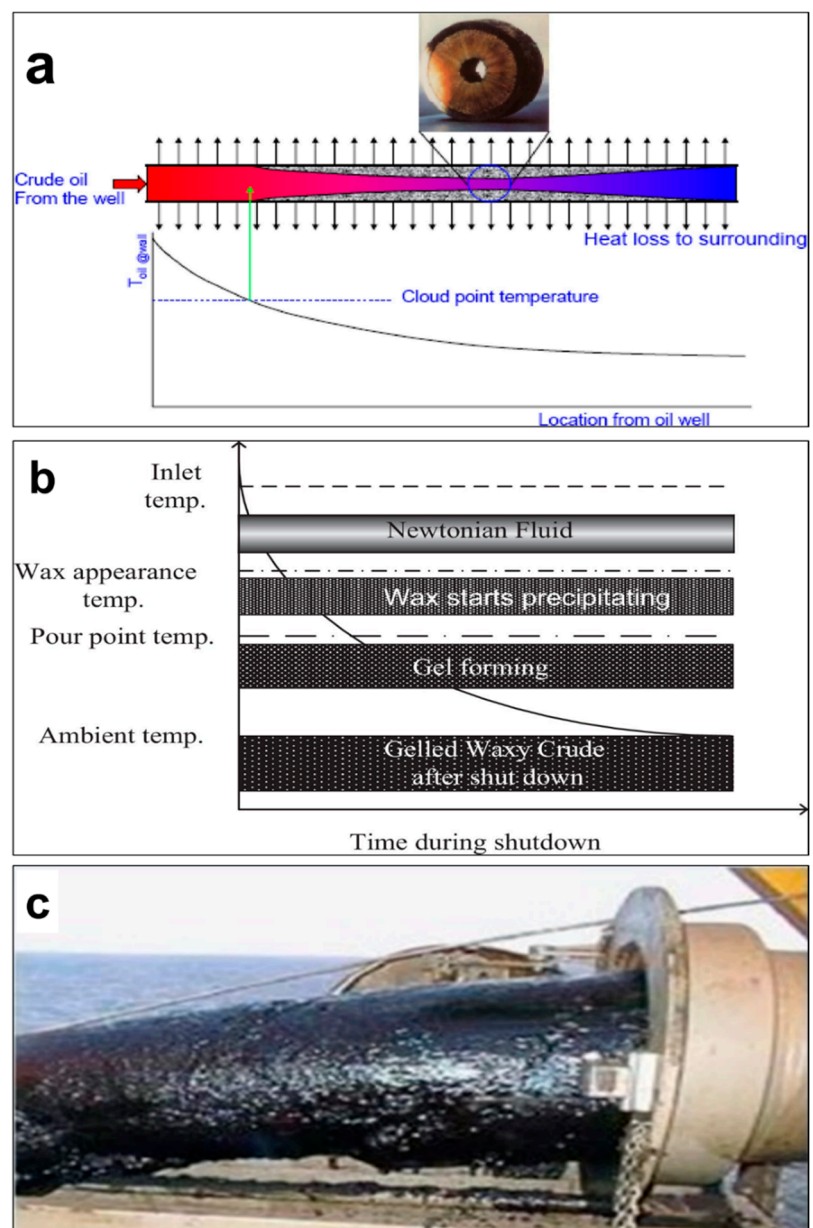

**Figure 4.** Occurrence of wax deposition once the internal temperature is less than the cloud point temperature (**a**), Development of gas holes in gelled waxy crude oil (**b**), and gelled crude oil in a pipe (**c**). Adapted with permission from [51], Elsevier, 2018.

A study by Kelechukwu et al. [132] was conducted to investigate the relationships between wax deposition in pipelines and changes in temperature, residence time, and flow rate. The flow loop consisting of the following components was used: a stainless steel tank that acted as the reservoir, identical pipes (twin) exchanger, a bath system with temperature controller, and a hydrostatic pump. The temperature was 40–55 °C, and the flow rate varied, with values in the 100–500 mL/min range [132].

When the temperature is below the WAT due to heat loss to the surroundings, the dissolved waxes from the liquid crude oil precipitate and are deposited on the pipeline walls (Figure 4b,c).

Temperature changes were investigated in another study using a flow loop setup that included a 5 L hot bath equipped with a crude oil sample. The crude oil was pushed into a test-tube within the flow line that was bounded by a jacket to circulate cold water continuously. Deposition was initiated when the pumped crude oil sample was cooled to the WAT ($-32\ ^\circ$C). The flow rate was calculated by evaluating the crude oil volume which collected over a fixed time. Flow was stopped after crude oil flowed into the test tube for a defined time interval. The test tube weight was determined afore and after the experiment, then the difference indicates the amount of deposited wax [133]. It was concluded that the temperature difference between the fluid within the pipeline and the wall of pipeline is the most important factor for wax deposition. Table 2 shows the pour point and wax appearance temperatures of waxy crude oil from different fields. Due to differing results produced by different studies, there is still uncertainty whether a huge temperature differential leads to less wax deposition. However, it is understood that the amount of deposited wax decreased with increased flow in both experimental setups. Further work is required in this area to obtain more concrete results so that a more accurate assessment of the influence of temperature upon wax deposition in subsea pipelines can be made.

### 4.2. Type of Crude Oil and Its Constituents

Crude oil content is one of the potential factors that influences on deposition of wax significantly and is liable for the viscosity and pour point reduction [13]. In addition, it contains small amounts of sulfur, nitrogen, and oxygen compounds [4]. Crude oils are categorized as either paraffin base, intermediate base, naphthenic base, or mixed base. Additional processing of crude oil involves separation of water, oil, and salt. Crude oil is brownish green to black in color and is a viscous mobile liquid, sometimes semisolid. The constituents of crude oil (including hydrocarbons and non-hydrocarbons) and their effect on wax formation in crude oil are summarized in Figure 4. The main cause of wax or paraffin deposition is a loss in solubility in the crude oil. This loss of solubility results from changes in composition, pressure, or temperature of the crude oil.

The cloud-point temperature (CPT) or wax appearance temperature increases due to the pressure effect on wax precipitation. Wax deposition from gas condensate fluids may have some unique features such as the retrograde phenomena for the precipitated solid phase. The phase behaviour of the wax enclosing condensate gas is complex [134]. At a constant temperature, the pressure drop may cause wax formation, which then vaporize or liquefies with continuous pressure decrease [135]. Therefore, ignoring wax precipitation will effect on the dew point pressure, as well, the Equation of state will be incorrect [136]. The solid wax is determined by long plate-like coarse wax crystals, when the pressure is above the dew point. However, when the pressure value is below the dew point, the solid wax is determined by condensate oil forming a glassy block material and transparent colloidal. The condensate gas does not accumulate wax in the wellbore during production process. However, measures should be evaluated to avoid wax formation and condensate generation caused by surface temperature and pressure changes [134].

## 5. Conclusions

Wax deposition is a common worldwide problem in the petroleum industry. At lower temperatures, heavy paraffin within reservoir fluids may deposit in solid form. The crystallization of wax is controlled by the temperature and the hydrocarbon composition. The amount of wax increases with high cooling time and solvent ratio. The formation of wax significantly influences crude oil properties, such as pour point and viscosity, and ultimately affects the flow conditions of the fluids, which further results in the formation of gel fluid. This causes serious issues, including the plugging of flow strings, the loss of hydrocarbons, and an increase in production costs. The ongoing research on detecting and minimizing of wax deposition has the prospective of easier crude oil maintenance

in the pipelines. Optimization of pigging frequency, and required pressure to restart gelled lines, is essential to prevent continuous wax removal processes. In terms of the computational modeling of the wax deposition phenomenon, it generally only takes the molecular diffusion into account as it is the most dominant mechanism. Recent studies are focusing on more complex environments, such as wax deposition behavior under multi-phase flow, turbulent flow, etc. It also should be noted that although the modeling results are verified with the lab-scale experimental approaches, there still are debates over its validity as the field scale observations suggest differences due to the scales.

**Author Contributions:** M.M.E.-D. & B.-H.J. designed the content of the manuscript and M.M.E.-D. wrote Section 1.3, Section 2.1, Section 3.1, Section 3.2 & Section 5; E.-S.S. wrote Section 1.1, Section 4.1 & Section 4.2; M.E. wrote Section 1.4; J.W. wrote Section 1.2; T.A. wrote Section 2.2, Section 2.3 & Section 2.4; W.B.K. & B.-H.J. improved the presentation of Figures and Tables; J.W., T.A., E.-S.S., M.M.E.-D. & B.-H.J. revised the manuscript; B.-H.J. & T.A. conceived and majorly coordinated the manuscript. All authors read and approved the final manuscript.

**Funding:** This research received no external funding.

**Acknowledgments:** This work was supported by Korea Institute of Energy Technology Evaluation and Planning (KETEP) (No. 20182510102400) and the Ministry of Trade, Industry & Energy (MOTIE) of the Republic of Korea (No. 20182510102400). This work was also supported by the research fund of Hanyang University (No. 201800000000288) and the project of KIGAM (GP2017-024).

**Conflicts of Interest:** The authors declare no conflict of interest.

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
