# Peer review of "Occurrence and Characterization of Paraffin Wax Formed in Developing Wells and Pipelines"

_energies, doi:10.3390/en12060967_

Round 1

Reviewer 1 Report

The review paper has provided a very good insight into the mechanism of wax formation, deposition and treatment methods in reservoir and pipelines, but will require additional critical arguments on the practical implications to the subject area through providing relevant evidence of production problems especially for the oil and gas industry  in terms of consequences of wax along pipelines and production tubing.

Specific comment:

Line 70: “To avoid various problems of plugging and clogging of pipelines” you will need to expand by giving examples of such relevant practical problems especially along the pipelines and oil production tubing. For example, reducing oil production from 24,000bpd to zero within 24 hrs in North Sea field etc .  You can access many of such practical examples through http://usir.salford.ac.uk/33069/

Line 125: Similar examples of inorganic scale growth along pipelines also needed to support this review as paraffin wax scale sometimes appear as a mixture with other inorganic scale in practical scenarios.

Line 158: Units of permeability and viscosity are required

Line 163: Table 3: Causes of formation damage has been tabulated, but require additional brief description on each of the stated cause. For example, CO2 flooding, could involve both miscible and immiscible interaction with the oil,  targeting reduction of the oil viscosity and wettability etc.

165: Paraffin wax management using pigging operations is one among many other operations especially for larger pipelines, but for smaller lines such as production tubing, the use of mechanical,  chemical methods, which the author has discussed, but will require examples to strengthen the critical review. This could include the use of high pressure water sprays as mechanical means, which later was followed by injection a mixture of sterling beads as conducted by Schlumberger Inc, to enhance erosion has been applied. Recent technique of using aeration-cavitation in descaling production tubing covered with wax and inorganic scale can also  be discussed here.

Line 175: Although chemical methods can be preferred, there is a need to critically discussed the environmental consequence of using chemicals, and the manner in which water-based chemicals and other environmentally friendly chemicals are employed to ensure environmental protection.

Author Response

February 19th, 2019

Editor: Energies

Response to review of manuscript (energies-433124R1)

Dear Prof. Lily Peng,

We would like to thank the Editor for providing a chance to our manuscript entitled “Occurrence and characterization of paraffin wax formed in developing wells and pipelines”, for possible publication. We would like to extend our gratitude towards the reviewers for their time and effort to review our manuscript; and appreciate the reviewer’s comments that would certainly improve the quality of the manuscript. The responses to the comments are provided point by point as raised by editor and reviewers. The modifications made in the revised manuscript were highlighted in red color. We are sure that this would satisfy reviewer’s concerns.

Please find below the responses to the corrections raised by the reviewers, along with the list of changes that we have made in the revised manuscript. The permission has been obtained for the use of copyrighted materials from other sources as the author guidelines of Energies. In addition to the changes suggested by the reviewers, we have also corrected some grammatical errors in the manuscript.

Reviewer #1:

The review paper has provided a very good insight into the mechanism of wax formation, deposition and treatment methods in reservoir and pipelines, but will require additional critical arguments on the practical implications to the subject area through providing relevant evidence of production problems especially for the oil and gas industry in terms of consequences of wax along pipelines and production tubing.

Reply to general comment: Authors appreciate the overall evaluation and suggestions for the manuscript. We have done our best to revise the manuscript to fully reflect the reviewer comments.

Critical discussion on the practical field has been incorporated in the revised manuscript (page 10, lines 288-300).

Specific comment:

1.                  Line 70: “To avoid various problems of plugging and clogging of pipelines” you will need to expand by giving examples of such relevant practical problems especially along the pipelines and oil production tubing. For example, reducing oil production from 24,000bpd to zero within 24 hrs in North Sea field etc.  You can access many of such practical examples through http://usir.salford.ac.uk/33069/

Reply to comment #1: Thank you for your helpful comment. More explanation and discussion from the recommended reference and available literature have been cited in the revised manuscript (page 7, lines 188-194).

2.                  Line 125: Similar examples of inorganic scale growth along pipelines also needed to support this review as paraffin wax scale sometimes appear as a mixture with other inorganic scale in practical scenarios.

Reply to comment #2: Thank you for your valuable comment. We have discussed in the revised manuscript the deposition of mixed scale (inorganic and organic) in the pipelines (page 10, lines 273-278).

Mixed scales are the inorganic and organic scales which occur simultaneously in the same system. Inorganic scales caused by calcium, barium, and strontium salts (CaCO3, CaSO4, BaSO4, and SrSO4) sometimes appear as a mixture with paraffin wax scale which create blockage over a period of time [1]. Mixed scales may result in highly complex structured scales that are difficult to treat. Such scales are formed when the pressure decreased in the tubing, CO2 released, and water evaporated, thus require aggressive, severe and sometimes costly remediation techniques [2]

[1]   Kumar, D.; Chishti, S. S.; Rai, A.; Patwardhan, S. D. In Scale inhibition using nano-silica particles, SPE Middle East Health, Safety, Security, and Environment Conference and Exhibition, 2012; Society of Petroleum Engineers: 2012.

[2]   Frenier, W. W.; Ziauddin, M., Formation, removal, and inhibition of inorganic scale in the oilfield environment. Society of Petroleum Engineers Richardson, TX: 2008.

3.                  Line 158: Units of permeability and viscosity are required

Reply to comment #3: The unit of permeability and viscosity has been incorporated in the revised manuscript (page 13, lines 366-367).

4.                  Line 163: Table 3: Causes of formation damage has been tabulated, but require additional brief description on each of the stated cause. For example, CO2 flooding, could involve both miscible and immiscible interaction with the oil, targeting reduction of the oil viscosity and wettability etc.

Reply to comment #4: Thank you for your valuable comment. Additional explanation has been incorporated in the revised manuscript (page 11, lines 310-335).

5.                  Line 165: Paraffin wax management using pigging operations is one among many other operations especially for larger pipelines, but for smaller lines such as production tubing, the use of mechanical,  chemical methods, which the author has discussed, but will require examples to strengthen the critical review. This could include the use of high pressure water sprays as mechanical means, which later was followed by injection a mixture of sterling beads as conducted by Schlumberger Inc., to enhance erosion has been applied. Recent technique of using aeration-cavitation in descaling production tubing covered with wax and inorganic scale can also be discussed here.

Reply to comment #5: Thank you for your comment. Discussion and examples on mechanical and chemical methods for paraffin wax management have been incorporated in the revised manuscript (page 14, lines 418-424).

6.                  Line 175: Although chemical methods can be preferred, there is a need to critically discuss the environmental consequence of using chemicals, and the manner in which water-based chemicals and other environmentally friendly chemicals are employed to ensure environmental protection.

Reply to comment #6: Thank you for your comment. Section 2.3 has been significantly improved in the revised manuscript with addition of critical discussion on the environmental consequence of using chemicals for paraffin control (page 15, lines 435-439).

We hope that we have addressed all reviewers’ comments sufficiently. Again, we greatly appreciate you for handling our manuscript and look forward to hearing from you regarding this submission of revised manuscript.

Yours sincerely,

Byong-Hun Jeon, Ph.D.

Reviewer 2 Report

There is an extensive literature which topic is to study the mechanisms of wax formation in pipelines and remedies. The paper needs a more extensive literature review The author should for example consider the following papers: 

Leporini M. et al, 2019, Experiences in numerical simulation of wax deposition in oil and multiphase pipelines: Theory versus reality, Journal of Petroleum Science and Engineering

pp. 997-1008

G. Giacchetta et al., Pipeline wax deposition modeling: A sensitivity study on two commercial software, Petroleum, https://doi.org/10.1016/j.petlm.2017.12.007

The authors should include a paragraph or at list a brief overview on the effects of the wax deposition on the flow assurance of hydrocarbon.

Author Response

February 19th, 2019

Editor: Energies

Response to review of manuscript (energies-433124R1)

Dear Prof. Lily Peng,

We would like to thank the Editor for providing a chance to our manuscript entitled “Occurrence and characterization of paraffin wax formed in developing wells and pipelines”, for possible publication. We would like to extend our gratitude towards the reviewers for their time and effort to review our manuscript; and appreciate the reviewer’s comments that would certainly improve the quality of the manuscript. The responses to the comments are provided point by point as raised by editor and reviewers. The modifications made in the revised manuscript were highlighted in red color. We are sure that this would satisfy reviewer’s concerns.

Please find below the responses to the corrections raised by the reviewers, along with the list of changes that we have made in the revised manuscript. The permission has been obtained for the use of copyrighted materials from other sources as the author guidelines of Energies. In addition to the changes suggested by the reviewers, we have also corrected some grammatical errors in the manuscript.

Reviewer #2:

There is an extensive literature which topic is to study the mechanisms of wax formation in pipelines and remedies. The paper needs a more extensive literature review. The author should for example consider the following papers: 

·                     Leporini M. et al, 2019, Experiences in numerical simulation of wax deposition in oil and multiphase pipelines: Theory versus reality, Journal of Petroleum Science and Engineering. pp. 997-1008

·                     G. Giacchetta et al., Pipeline wax deposition modeling: A sensitivity study on two commercial software, Petroleum, https://doi.org/10.1016/j.petlm.2017.12.007

The authors should include a paragraph or at list a brief overview on the effects of the wax deposition on the flow assurance of hydrocarbon.

Reply to the general comment: Thank you so for you evaluation and suggestions for the manuscript. We have done our best to revise the manuscript to fully reflect the reviewer comments.

The manuscript has been thoroughly revised with addition of more literature. Moreover, a new sub-section entitled “2.3. Effects of wax deposition on the flow assurance of hydrocarbon” has been incorporated. The suggested references, to address the effects of wax deposition on the flow assurance of hydrocarbon, have been cited in the revised manuscript (page 13, lines 370-384).

We hope that we have addressed all reviewers’ comments sufficiently. Again, we greatly appreciate you for handling our manuscript and look forward to hearing from you regarding this submission of revised manuscript.

Yours sincerely,

Byong-Hun Jeon, Ph.D.

Reviewer 3 Report

In the manuscript "Occurrence and characterization of paraffin wax formed in developing wells and pipelines", the authors El-Dalatony and co-workers review the phenomenon of wax deposition in reservoirs and tubing. The title of the manuscript does not reflect the contents of the paper, and should be broadened to also include reservoir phenomenon.

The most important change required in the manuscript is to include the conditions of wax deposition, that the depositing surface should be colder than the WAT, and there must be a negative radial thermal gradient in the pipeline, or an analogous mathematical difference in the temperature (and by inferrence, solubility) of the waxy reservoir oil and the reservoir rocks.

Major concerns:

1. Each composition percentage value in the entire manuscript must be quality assured, contexualized and appropriately cited. For example, the 32.5% value of wax content listed on the first page, line 39, is incorrect, and many of the other compositional values in the manuscript are incorrect as well. We cannot approve a manuscript with incorrect compositional values. The wax content varies from field to field more than from country to country. Also, the composition of napthenatic acids, sulfur components, and etc. varies from field to field, and even varies within portions of field reservoirs, given a lack of flow communication between internal zones of specific field reservoirs. Hence, all of the compositional values given in the paper must be put into proper context and substantially improved. We understand that this is an ethical requirement for publication in MDPI Energies.

2. Wax deposition will depend on the surface chemistry of the reservoir rock (if occurring in a reservoir) or the surface chemistry of the inside wall of the tubing. Metallic surfaces typically have an oxide layer at the outer surface that attracts an adsorbed water film that hinders the incipient deposition of wax on the surface. The micro-crystalline wax can more easily overcome this hindrance and deposit despite the hydrophilic resistance. The macro-crystalline types of wax are more hindered by the adsorbed water film. Similarly, if fluorinated surfaces are used, the electronegativity of the fluorine atom holds the electrons in the SP3 hybridized orbital very tight, which minimized the Debye van der Waals forces (permanent-induced) and the London van der Waals forces (induced-induced) with the paraffin wax molecules. The Keesom forces are not relevant due to the non-polar nature of the paraffin wax molecules. Hence, fluorinated surfaces minimize the attractive forces between the surface and the wax molecules, whether in soluble or crystallized form, and provide a greater resistance to incipient wax deposition than the metal oxide surfaces with the adsorbed water film. The strong implication of the surface chemistry is that fluorinated surfaces (including fluorinated coating products from DuPont, etc.) are able to in many cases completely prevent wax deposition on solid surfaces. Field experience from the Americas has shown that fluorinated coatings have been able to completely prevent wax deposition in wells. Usually the fluorinated surface will be able to prevent the depsition of wax up to a given compositional driving force, established on a solubility difference threshold. We sugest that the authors include relevant discussions about the influence of surface chemistry of the deposition surface on the incipient wax deposition. Please see the following book chapter: Encyclopedia of Surface and Colloid Science: Wax Deposition Prevention by Surface Modifications: Materials Evaluation. CRC Press. 2012. Also please see the following book chapter:  Handbook of Surface and Colloid Chemistry: Wax Deposition Investigations with Thermal Gradient Quartz Crystal Microbalance. 2008.

3. The two critical carbon numbers (CCN) should be signaled. The "lower critical carbon number" and the "upper critical carbon number" establish on a compositional basis the range of components that contribute to the deposition process, respectively. If the deposition surface temperature is lower than the bulk fluid temperature and the deposition surface temperature is less than the WAT, then the "lower critical carbon number" will establish a lower bound to the range of components that contribute to the deposition process. If the bulk fluid temperature is below the WAT and the deposition surface temperature is below the WAT, then the "upper critical carbon number" establishes an upper limit to the range of components that contribute to the deposition process. There is substantial literature avialable on the compositional basis of wax deposition.

4. Mathematical modelling of wax deposition using convective heat and mass transfer correlations must be signalled, considering that the scope of the title includes the occurence of paraffin wax formation. There exists an abundance of literature available on this topic using a simple google search.

5. HTGC methods for characterization of the wax should be signalled. These methods are very commonly used in industry. It is very important to signal HTGC because this is the most common method and easiest method to characterize the wax itself. Most paraffinic crude oils can be injected directly into the HTGC without prior separation of the asphaltenes, because the aphaltenes do not have a vapor pressure. However, even after separation of the asphaltenes, the HTGC method can still be used. The FID is the most common detector for HTGC. New 2D-HTGC techniques may also be useful for understanding the structure of the wax molecules. Other methods for establishing the number of rings on the wax structure can also be signalled in the manuscript.

6. Because the WAT determines where the wax will deposit, measurement methods for the WAT must be signalled in the manuscript. This is very important for a practical perspective for the readership of MDPI Energies journal.

7.  Paraffin wax can cause problems with wax deposition as well as with plugging of pipelines after flow shutin for planned maintenance or emergency contingency situations. Hence, this related problem of wax gelling in the pipeline after shut-in and pipeline restart after gelling should be briefly mentioned in the manuscript, because several of the analysis techniques and scientific/engineering considerations are identical for both related industrial problems. In fact, it can be shown by TG-QCM that the incipient deposition mechanism is actually gelation. (Please see the book chapter above on the Thermal Gradient Quartz Crystal Microbalance)

8. Wax deposition in gas condensate fluids must be signalled. These are not crude oils, but can have just as many problems with deposition in the wellstring and in transport pipelines as crude oils. The wax content in gas condensate fluids can very very high, and should be signalled to the readership of MDPI Energies.

Specific line comments:

Line 36. Not all petroluem fluids contain wax, so line 36-37 should be modified.

Line 41. Oil gelatination should be gelation. Flow tendencies should be flowability

Line 42. Waxes are non-polar, not polar. Wax itself does not contain aromatic/napthenic hydrocarbons, regardless of MW.

Line 64. The statement "Thus, micro-crystalline wax has a higher tendency to form stiff gel than macro-crystalline wax" is highly misleading. The micro-crystalline wax usually crystallizes at a higher temperature, but, so at high T the micro-crystalline wax may form stronger gels, but a low temperatures the macro-components usally form stronger gels at low solid fractions, due to percolation considerations and the fact that micro-components are generally lower in composition.

Line 67. Propylene cannot be described as having one pair of carbon atoms. Also, NM is unacceptable for an MDPI journal. Also, combustion properties are not within the scope of the manuscript title.

Line 91-92. This is incorrect.

Line 98. This is incorrect.

Line 102. The acetone precipitation method for wax must be signalled. The spectroscopic methods are not used very often for wax content determination, and are most likely unreliable.

Line 134. "When the temperature drops below the pour point, deposition of paraffin crystals is initiated" This statement is simply not true. The PP has no relation to the deposition onset. Please see the accepted conditions for wax deposition provided in the following book: Wax Deposition: Experimental Characterizations, Theoretical Modelling, and Field Practices, First Edition, Authors: Zhenyu Huang, Sheng Zheng, H. Scott Fogler. The conditions for wax deposition are exactly the same in the pipeline and in the reservoir, with the reservoir bulk fluid representing the bulk fluid in the pipeline.

Line 137-144. It should be specifically mentioned that hot oiling dissolves paraffin components, increasing the wax content, and then subsequent cooling of the initial hot oil causes subsequent wax deposition.

Line 148. Agglutination is not a word.

Line 180. We agree that chemical methods are difficult to standardize.

Line 205. The acetone precipitation method is much more common than ether/ethanol precipitation method. The -20 degrees Celsius specification is correct.

Line 247-256. Please see the major concern #3 above.

Line 262. "The WAT of micro-crystalline waxes is 4 to 6 times higher than that of macro-crystalline waxes" Generally, this statement is incorrect and should be removed. It is highly misleading.

Line 284. We like the images. Excellent!

Line 296-300. We agree with this statement. This sub-field warrants additional research.

Line 305. This given window for carbon and hydrogen is too narrow and thereby incorrect, and in any case must be qualified and contextualized.

Line 317-318. "The amount of wax increases with increased initial crude weight" This statement is incorrect and misleading. Heavy crude oil and extra heavy crude oils often times have no wax or very little wax.

In conclusion, the authors have an incorrect understanding of wax deposition on many levels, and the manuscript is highly misleading in many ways.

The authors must perform a more comprehensive literature review. A good starting point for a comprehensive literature review is the book by Zhenyu Huang listed above, which happens to be the only book entirely about wax deposition, even though some other book chapters exist that are devouted to wax deposition. Also, a more comprehensive literature review of WAT, HTGC, and anti-wax coatings is warranted.

In current form, this manuscript is a disservice to the scientific community and should be completely rejected due to the abundance of misleading statements, especially concerning compositional parameters, which are known to vary widely from field to field. Our true feeling is that it would be somewhat unethical to publish this manuscript in current form, due to the abundance of misleading statements.

We ask that this manuscript is completely rejected, and then subsequently re-submitted by the authors after a more comprehensive literature review is performed.

We feel that it is very important to give the authors a fresh start by rejecting and resubmitting. In this way the authors can choose an overall better structure for the manuscript, and will not be burdened by having to referring back to an irrelevant version of the manuscript. For example, the authors can completely discard the unrelated sections concerning combustion, refined petroleum derivatives ethylene/propolyene, and etc.

(Also this approach will give the authors more flexibility in elevating the grammatical level of the manuscript)

Author Response

February 19th, 2019

Editor: Energies

Response to review of manuscript (energies-433124R1)

Dear Prof. Lily Peng,

We would like to thank the Editor for providing a chance to our manuscript entitled “Occurrence and characterization of paraffin wax formed in developing wells and pipelines”, for possible publication. We would like to extend our gratitude towards the reviewers for their time and effort to review our manuscript; and appreciate the reviewer’s comments that would certainly improve the quality of the manuscript. The responses to the comments are provided point by point as raised by editor and reviewers. The modifications made in the revised manuscript were highlighted in red color. We are sure that this would satisfy reviewer’s concerns.

Please find below the responses to the corrections raised by the reviewers, along with the list of changes that we have made in the revised manuscript. The permission has been obtained for the use of copyrighted materials from other sources as the author guidelines of Energies. In addition to the changes suggested by the reviewers, we have also corrected some grammatical errors in the manuscript.

Reviewer # 3

In the manuscript "Occurrence and characterization of paraffin wax formed in developing wells and pipelines", the authors El-Dalatony and co-workers review the phenomenon of wax deposition in reservoirs and tubing. The title of the manuscript does not reflect the contents of the paper, and should be broadened to also include reservoir phenomenon.

The most important change required in the manuscript is to include the conditions of wax deposition, that the depositing surface should be colder than the WAT, and there must be a negative radial thermal gradient in the pipeline, or an analogous mathematical difference in the temperature (and by inference, solubility) of the waxy reservoir oil and the reservoir rocks.

Reply to general comment: Thank you for your helpful comment. A new sub-section “1.1. Wax deposition” has been added in the revised manuscript to cover the conditions of wax deposition (page 2, lines 55-83).

Major concerns:

1.                  Each composition percentage value in the entire manuscript must be quality assured, contextualized and appropriately cited. For example, the 32.5% value of wax content listed on the first page, line 39, is incorrect, and many of the other compositional values in the manuscript are incorrect as well. We cannot approve a manuscript with incorrect compositional values. The wax content varies from field to field more than from country to country. Also, the composition of napthenatic acids, sulfur components, and etc. varies from field to field, and even varies within portions of field reservoirs, given a lack of flow communication between internal zones of specific field reservoirs. Hence, all of the compositional values given in the paper must be put into proper context and substantially improved. We understand that this is an ethical requirement for publication in MDPI Energies.

Reply to the comment #1: We agree with reviewer’s comment. Wax content may vary not only in the region but also in the same reservoir. Therefore, it is necessary to estimates it in a specific area, reservoir, or field. We have checked and corrected the compositional values in each part in the revised manuscript (lines 39 and 460).

2.                  Wax deposition will depend on the surface chemistry of the reservoir rock (if occurring in a reservoir) or the surface chemistry of the inside wall of the tubing. Metallic surfaces typically have an oxide layer at the outer surface that attracts an adsorbed water film that hinders the incipient deposition of wax on the surface. The micro-crystalline wax can more easily overcome this hindrance and deposit despite the hydrophilic resistance. The macro-crystalline types of wax are more hindered by the adsorbed water film. Similarly, if fluorinated surfaces are used, the electronegativity of the fluorine atom holds the electrons in the SP3 hybridized orbital very tight, which minimized the Debye van der Waals forces (permanent-induced) and the London van der Waals forces (induced-induced) with the paraffin wax molecules. The Keesom forces are not relevant due to the non-polar nature of the paraffin wax molecules. Hence, fluorinated surfaces minimize the attractive forces between the surface and the wax molecules, whether in soluble or crystallized form, and provide a greater resistance to incipient wax deposition than the metal oxide surfaces with the adsorbed water film. The strong implication of the surface chemistry is that fluorinated surfaces (including fluorinated coating products from DuPont, etc.) are able to in many cases completely prevent wax deposition on solid surfaces. Field experience from the Americas has shown that fluorinated coatings have been able to completely prevent wax deposition in wells. Usually the fluorinated surface will be able to prevent the deposition of wax up to a given compositional driving force, established on a solubility difference threshold. We suggest that the authors include relevant discussions about the influence of surface chemistry of the deposition surface on the incipient wax deposition. Please see the following book chapter: Encyclopedia of Surface and Colloid Science: Wax Deposition Prevention by Surface Modifications: Materials Evaluation. CRC Press. 2012. Also please see the following book chapter:  Handbook of Surface and Colloid Chemistry: Wax Deposition Investigations with Thermal Gradient Quartz Crystal Microbalance. 2008.

Replay to the comment #2:  Thanks to the reviewer for providing us this very important piece of information. Relevant discussions about the effect of surface chemistry of the reservoir rock, wall of the tubing, and composition of crude-oil have been incorporated in revised manuscript. Authors considered the recommended references by the reviewer and the available literature. Several curial factors (including surface chemistry of reservoir rock, inside wall of the tubing, and crude-oil composition) influence the incipient wax deposition. Therefore, several studies have been performed to understand the interactions among composition of reservoir rocks (such carbonate rock or/and sandstone), tubing wall materials (metallic, ceramic or micanite), composition of crude-oil (organic acids, salts, alcohols, and other natural surface-active agents), as well as types of produced wax (macro-crystalline wax and micro-crystalline wax). Such studies need be discussed for improvement of the oil recovery from the reservoir.

Two major factors have been reported to affect the wettability: 1) rocks surface morphology and 2) the intermolecular surface forces among the three phases (i.e., rock, oil and brine). Irrespective of the morphology, the wettability of the system is determined by the relative magnitude of the forces of interaction between the two liquid phases and the rock surface. The rock reservoirs were commonly categorized as oil-wet, water-wet or intermediate-wet based on the affinity of the rock surface toward oil or water phase [1]. The wettability of the system is determined by the relative magnitude of the forces of interaction between the two liquid phases and the rock surface. Such interactions and surface energies are usually classified into two classes: a) Lifshitz-van der Waals interactions (non-polar) and b) acid-base interactions (polar). Therefore, there are two indirect approaches usually used to evaluate the surface energy of solids (vapor adsorption measurements via probe vapors) and wetting (contact angle) measurements using probe liquids. Arsalan et al. [1] established a new method to quantify these fundamental interactions through characterization of the surface energetics of carbonate rocks some and sandstone by inverse gas chromatography.

A surfactant molecule has two functional groups, namely a hydrophilic (water-soluble) or polar group and a hydrophobic (oil-soluble) or non-polar group. The hydrophobic group is usually a long hydrocarbon chain (C8-C18), which may or may not be branched, while the hydrophilic group is formed by moieties such as carboxylates, sulfates, sulfonates (anionic), alcohols, polyoxyethylenated chains (nonionic) and quaternary ammonium salts (cationic) [2]. Crude-oil contains organic acids and salts, alcohols and other natural surface-active agents. When crude oil is brought in contact with brine or water, these natural surfactants accumulate at the interface and form an adsorbed film which lowers the interfacial tension of the crude oil/water interface. Depending on the type of crude oil, the adsorbed film at the interface can be either fluid or very viscoelastic and able to form a skin [3]. Therefore, the molecular packing, surface viscosity, surface elasticity, and surface charge of the adsorbed film are very vital parameters that determine various phenomena such as coalescence of emulsion droplets, as well as oil drop migration in porous media. It has been reported that, ceramic or micanite in-line ring heaters are recommended to be applied for prevention of wax accumulation in Tubing (Fig. 1). Apart of this discussion has been added in the revised manuscript (page 2, lines 69-83) and (page 15, lines 445-457).

Fig 1. Suggest materials to be used for creating the required temperature in the very wax accumulation zone: Ring heaters are ideal for heating tubes and cylinders (a) and energy-conserving ring clamp heater (b) [4].

References:

[1]   Arsalan, N.; Buiting, J. J.; Nguyen, Q. P. J. C.; Physicochemical, S. A.; Aspects, E., Surface energy and wetting behavior of reservoir rocks. 2015, 467, 107-112.

[2]   Paso, K.; Viitala, T.; Aske, N.; Sjöblom, J., Wax deposition prevention by surface modifications: materials evaluation. Encyclopedia of Surface and Colloid Science (second edition), CRC Press, Boca Raton 2012.

[3]   Kanicky, J. R.; Lopez-Montilla, J.-C.; Pandey, S.; Shah, D. O., Surface chemistry in the petroleum industry. Handbook of applied surface and colloid chemistry 2001, 1, 251-267.

[4]   Musipov, H.; Akhmadulin, R.; Bakanovskaya, L. In Wax Accumulation Prevention Method in Tubing, IOP Conference Series: Materials Science and Engineering, 2017; IOP Publishing: 2017; p 012018.

3.                  The two critical carbon numbers (CCN) should be signaled. The "lower critical carbon number" and the "upper critical carbon number" establish on a compositional basis the range of components that contribute to the deposition process, respectively. If the deposition surface temperature is lower than the bulk fluid temperature and the deposition surface temperature is less than the WAT, then the "lower critical carbon number" will establish a lower bound to the range of components that contribute to the deposition process. If the bulk fluid temperature is below the WAT and the deposition surface temperature is below the WAT, then the "upper critical carbon number" establishes an upper limit to the range of components that contribute to the deposition process. There is substantial literature available on the compositional basis of wax deposition.

Replay to the comment #3:  We agree with the reviewer in that, the discussion on lower critical carbon number and upper critical carbon number along with deposition surface temperature should be signaled in the manuscript. Such information has been now cleared in the revised manuscript (pages 13-14, lines 385-397). Wax deposition is effected by temperature and critical carbon number (CCN) (Fig. 2). The CCN has been reported to be C27-C28. Wax deposition occur when the deposition surface temperature is below than both the bulk fluid temperature and the wax appearance temperature (WAT).  The lower carbon number (<27) start to deposit on the pipeline surface when deposition surface temperature is lower than bulk fluid (i.e., crude-oil temperature) and WAT, resulting in micro-crystallization. Future decrease in surface deposition temperature cause macro-crystallization of upper carbon number (>28) replacing micro-crystallization. Wax molecules with upper carbon number than CCN diffused into the deposit, while those with carbon number less than CCN diffused out from the deposit and increases the hardness of deposit (this process is called “aging” or “nucleation”) [1]. Initially, the rate of wax deposition is high, but it slows down as more wax is deposited on the pipe surface. The thickness of the wax layer (macro- crystallization) increases and acts as thermal insulation factor. The wax deposition and its thickness control by several mechanism such as Shear dispersion, Brownian diffusion, gravity settling, and molecular diffusion [2].

Fig. 2. Changes in carbon number distribution of deposit with time (Toil=WAT+ 5 °C, TCoolant= WAT -10 °C) [3].

References

[1] Huang, Z.; Lee, H. S.; Senra, M.; Scott Fogler, H. J. A. J., A fundamental model of wax deposition in subsea oil pipelines. 2011, 57, (11), 2955-2964.

[2] Roehner, R.; Fletcher, J.; Hanson, F.; Dahdah, N. J. E.; fuels, Comparative compositional study of crude oil solids from the Trans Alaska Pipeline System using high-temperature gas chromatography. 2002, 16, (1), 211-217.

[3] Quan, Q.; Gong, J.; Wang, W.; Gao, G., Study on the aging and critical carbon number of wax deposition with temperature for crude oils. Journal of Petroleum Science and Engineering 2015, 130, 1-5.

4.      Mathematical modelling of wax deposition using convective heat and mass transfer correlations must be signaled, considering that the scope of the title includes the occurrence of paraffin wax formation. There exists an abundance of literature available on this topic using a simple google search.

Replay to the comment #4:  Thank you for your helpful comment. We have added a new sub-section entitled “1.2.    Mathematical modelling of wax deposition” in the revised manuscript (pages 3-4, lines 85-163). This sub-section includes the following information:

There are four wax deposition mechanisms need to be taken into account for modeling the wax deposit growth in pipelines, i.e. molecular diffusion, shear dispersion, gravity settlement and Brownian diffusion[1, 2]. The molecular diffusion, which can be described by taking the mass transfer and energy balance into account, is considered as the dominant mechanism during the wax deposition process [1, 3-6]. Burger, Perkins and Striegler [1] and Weingarten and Euchner [7] proposed that the shear dispersion mechanism needs to be incorporated for the wax deposition behavior analysis, especially in laminar flow. However, Brown, Niesen and Erickson [4] performed a set of experiments and pointed out that the mechanism does not take a role during the wax deposition. The gravity settling mechanism suggests that the wax crystals are would be precipitated to the bottom of the pipelines as the crystals are denser than the oil. However, it appears that the effect is negligible as experimental studies comparing vertical and horizontal fluid flows did not reveal any difference in the deposited wax amount [8]. As soon as the wax crystals are precipitated and suspended in the oil, the wax crystals will behave with Brownian motion. Since the motion effect is likely to transport the crystals to the area with lower wax concentration, it may have an influence on the wax deposition behavior. However, its effect is generally considered minimal and it is generally neglected during modeling the overall deposition mechanisms [8]. Consequently, the latter three mechanisms are widely not accepted, especially for the computational modeling procedures [8-14]. In this part of the study, the most dominant mechanism, the molecular diffusion, is briefly reviewed.

In order to incorporate the molecular dispersion mechanisms, the radial temperature gradient of the pipeline needs to be taken into account. This is because a radial convective flux of wax molecules is induced by the concentration gradients, which strongly depends on the temperature gradient inside the pipeline. Therefore, the flux can be calculated by difference in the wax concentration between the bulk and the interface of the oil and the wax deposits. The mass transfer can be expressed with the equation as follows,

where,  is the clean pipe radius,  is the flowable radius,  is weight fraction of solid in the wax deposit,  is the pipeline length,  is the wax deposit density,  is the convective mass transfer coefficient,  and  are the dissolved paraffin concentration in the bulk and at the oil-gel interface, respectively.

Another process that needs to be incorporated is the growth of the deposited wax. Since there exists a temperature gradient within the deposit, an internal mass transfer also occurs. The wax deposition growth can be described by the equation as follows,

where,  is effective diffusivity of wax in the waxy given by Cussler, et al. [15]:

where,  is the molecular diffusivity of wax in oil and  is the aspect ratio of the wax crystals in the deposit.

Within the system the energy balance is met as the heat conduction across the deposit thickness is the sum of the radial convective heat flux and the latent heat of solidification. Therefore, the energy balance can be expressed with the equation as follows,

where, is the temperature of oil at the center,  is the oil-wax interface temperature,  is the ambient temperature, is the conductive heat coefficient,  is the inner convective heat transfer coefficient and  is the heat of wax solidification.

Assuming flow in the pipe is laminar, the heat transfer coefficients can be determined by the empirical correlations, such as the Hausen correlation [16] and the Seider and Tate correlation [17]. The former is valid for long tubes, while the latter shows a good agreement with a short tube length [6]. The Nusselt number for the Graetz number lower than 100 can be calculated by the Hausen correlation as follows,

where,  and  are Graetz number for heat transfer and for mass transfer, respectively. When the Graetz number is larger than 100, the Nusselt number can be obtained by the Seider and Tate correlation as follow,

The heat transfer coefficient and the mass transfer coefficient can be then calculated by the equations as follows,

where,  is the thermal conductivity of the oil.

The computational modeling can be a powerful tool to predict and to interpret the wax deposition phenomena after validated with available experiments data. However, most of the lab-scale experiments are based on the single-phase flow, which frequently yields deviated outcomes when compared to more-general multiphase flow cases. In addition, Leporini et al. [17] emphasized that the modeling results are not always applicable for the field scale analysis and that the scale effect also needs to be taken into account for more reliable interpretation. Consequently, recent studies focus on more complex environments, such as wax deposition behavior under multi-phase flow and turbulent flow.

References:

[1]         Burger, E. D.; Perkins, T. K.; Striegler, J. H., Studies of Wax Deposition in the Trans Alaska Pipeline. SPE-8788-PA 1981, 33, (06), 1075-1086.

[2]         Todi, S. Experimental and modeling studies of wax deposition in crude oil carrying pipelines. University of Utah, Utah, 2005.

[3]         Bern, P. A.; Withers, V. R.; Cairns, R. J. R., Wax Deposition in Crude Oil Pipelines. In European Offshore Technology Conference and Exhibition, Society of Petroleum Engineers: London, United Kingdom, 1980; p 8.

[4]         Brown, T. S.; Niesen, V. G.; Erickson, D. D., Measurement and Prediction of the Kinetics of Paraffin Deposition. In SPE Annual Technical Conference and Exhibition, Society of Petroleum Engineers: Houston, Texas, 1993; p 16.

[5]         Venkatesan, R.; Fogler, H. S., Comments on analogies for correlated heat and mass transfer in turbulent flow. AIChE Journal 2004, 50, (7), 1623-1626.

[6]         Singh, P.; Venkatesan, R.; Fogler, H. S.; Nagarajan, N., Formation and aging of incipient thin film wax-oil gels. AIChE Journal 2000, 46, (5), 1059-1074.

[7]         Weingarten, J. S.; Euchner, J. A., Methods for Predicting Wax Precipitation and Deposition. SPE-15654-PA 1988, 3, (01), 121-126.

[8]         Guo, B.; Song, S.; Ghalambor, A.; Lin, T. R., Chapter 15 - Flow Assurance. In Offshore Pipelines (Second Edition), Guo, B.; Song, S.; Ghalambor, A.; Lin, T. R., Eds. Gulf Professional Publishing: Boston, 2014; pp 179-231.

[9]         Lee, H. S. COMPUTATIONAL AND RHEOLOGICAL STUDY OF WAX DEPOSITION AND GELATION IN SUBSEA PIPELINES. University of Michigan, 2008.

[10]     Zheng, S.; Saidoun, M.; Mateen, K.; Palermo, T.; Ren, Y.; Fogler, H. S., Wax Deposition Modeling with Considerations of Non-Newtonian Fluid Characteristics. In Offshore Technology Conference, Offshore Technology Conference: Houston, Texas, USA, 2016; p 18.

[11]     Leporini, M.; Terenzi, A.; Marchetti, B.; Giacchetta, G.; Corvaro, F., Experiences in numerical simulation of wax deposition in oil and multiphase pipelines: Theory versus reality. Journal of Petroleum Science and Engineering 2019, 174, 997-1008.

[12]     Giacchetta, G.; Marchetti, B.; Leporini, M.; Terenzi, A.; Dall’Acqua, D.; Capece, L.; Cocci Grifoni, R., Pipeline wax deposition modeling: A sensitivity study on two commercial software. Petroleum 2017.

[13]     Li, S.; Huang, Q.; Zhao, D.; Lv, Z., Relation of heat and mass transfer in wax diffusion in an emulsion of water and waxy crude oil under static condition. Experimental Thermal and Fluid Science 2018, 99, 1-12.

[14]     Soedarmo, A. A.; Daraboina, N.; Sarica, C., Validation of wax deposition models with recent laboratory scale flow loop experimental data. Journal of Petroleum Science and Engineering 2017, 149, 351-366.

[15]     Cussler, E. L.; Hughes, S. E.; Ward, W. J.; Aris, R., Barrier membranes. Journal of Membrane Science 1988, 38, (2), 161-174.

[16]     Hausen, H., Darstellung des warmeuberganges in rohren durch verallgemeinerte potenz-beziehungen. ZVDI Beih Verfahrenstech 1943, 4, 91.

[17]     Sieder, E. N.; Tate, G. E., Heat Transfer and Pressure Drop of Liquids in Tubes. Industrial & Engineering Chemistry 1936, 28, (12), 1429-1435.

5.                  HTGC methods for characterization of the wax should be signaled. These methods are very commonly used in industry. It is very important to signal HTGC because this is the most common method and easiest method to characterize the wax itself. Most paraffinic crude oils can be injected directly into the HTGC without prior separation of the asphaltenes, because the asphaltenes do not have a vapor pressure. However, even after separation of the asphaltenes, the HTGC method can still be used. The FID is the most common detector for HTGC. New 2D-HTGC techniques may also be useful for understanding the structure of the wax molecules. Other methods for establishing the number of rings on the wax structure can also be signaled in the manuscript.

Replay to the comment #5:  We agree with reviewer’s comments that high temperature gas chromatography (HTGC) is one of the most effective method for wax determination. The distinction between general GC and HTGC is not clearly defined, but GC at 340 degrees or more can be called HTGC. HTGC enables the analysis of a wide range of molecular weights from C7 to > C100 [1]. Recently, two-dimensional GC with flame ionization detection (HT-GCxGC-FID) or a time-of-flight mass spectrometer (HT-GCxGC-TOFMS) has been used to separate n-paraffin, iso-paraffin, and cyclo-praffin [2]. Such information has been incorporated in the revised manuscript (page 15, lines 461-467).

References:

[1]   França, D.; Pereira, V. B.; Coutinho, D. M.; Ainstein, L. M.; Azevedo, D. A., Speciation and quantification of high molecular weight paraffins in Brazilian whole crude oils using high-temperature comprehensive two-dimensional gas chromatography. Fuel 2018, 234, 1154-1164.

[2]   Mahé, L.; Courtiade, M.; Dartiguelongue, C.; Ponthus, J.; Souchon, V.; Thiébaut, D., Overcoming the high-temperature two-dimensional gas chromatography limits to elute heavy compounds. Journal of Chromatography A. 2012, 1229, 298-301.

6.                  Because the WAT determines where the wax will deposit, measurement methods for the WAT must be signaled in the manuscript. This is very important for a practical perspective for the readership of MDPI Energies journal.

Replay to the comment #6:  Thank you for your valuable comment. As recommended by reviewer, brief explanations of industry established techniques for measuring WAT has been incorporated in the revised manuscript as the following:

WAT is the temperature at which wax crystal formation begins. The formation of wax causes the fluid to become cloudy, hence WAT is also called cloud point. The industry established techniques for measuring WAT in the laboratory are ranging from ASTM D2500, cross polar microscopy (CPM), light transmission (LT), differential scanning calorimetry (DSC), viscometry, and filter plugging infrared (FTIR) to cold finger test [1, 2]. The resultant values of WAT depend on the measurement techniques. The ASTM method relies on the visual observation of wax crystals, which reduces the reliability of WAT measurements of opaque or dark oils. Therefore, CPM, LT, DSC, or viscometry methods are preferred for WAT measurements of dark oils. DSC and viscometry are affected by the amount of precipitated wax, while light transmittance (LT) and light scattering are affected by the number of wax crystals. WAT measurement methods have advantages and disadvantages, so it is recommended to use multiple technologies together depending on the given conditions and circumstances [3, 4]. Such explanation has been incorporated in the revised manuscript (page 7, lines 197-208).

References:

[1]   Uba, E.; Ikeji, K.; Onyekonwu, M. In Measurement of wax appearance temperature of an offshore live crude oil using laboratory light transmission method, Nigeria Annual International Conference and Exhibition, 2004; Society of Petroleum Engineers: 2004.

[2]   Mmata, B.; Onyekonwu, M. In Measurement of the Wax Appearance Temperature of a Gas Condensate Using High Pressure Microscopy Technique, SPE Nigeria Annual International Conference and Exhibition, 2018; Society of Petroleum Engineers: 2018.

[3]   Monger-McClure, T.; Tackett, J.; Merrill, L., Comparisons of cloud point measurement and paraffin prediction methods. SPE production & facilities 1999, 14, (01), 4-16.

7.                  Paraffin wax can cause problems with wax deposition as well as with plugging of pipelines after flow shut-in for planned maintenance or emergency contingency situations. Hence, this related problem of wax gelling in the pipeline after shut-in and pipeline restart after gelling should be briefly mentioned in the manuscript, because several of the analysis techniques and scientific/engineering considerations are identical for both related industrial problems. In fact, it can be shown by TG-QCM that the incipient deposition mechanism is actually gelation. (Please see the book chapter above on the Thermal Gradient Quartz Crystal Microbalance).

Replay to the comment #8:  Thank you for your helpful comment. The related problem of wax gelling in the pipeline after gelling has been discussed in the manuscript (page 14, lines 398-411). We have added a new sub-section entitled 2.3. Effects of wax deposition on the flow assurance of hydrocarbon” in the revised manuscript to deal with flow assurance problems in detail. The pipeline restart problem by wax gelling has been incorporated in the last paragraph of the sub-section as the following:

Wax precipitation lead to wax deposition and flow restriction during the oil flow, which consequently cause wax precipitation during production also causes problems when trying to restart the flow [1-5]. If oil production is halted by a scheduled maintenance or an emergency situation, the temperature and solubility of wax decreases in a static condition. In this circumstances, wax-oil gel is formed due to precipitation of wax molecules, which can lead to blockage of the entire pipeline. In order to restart the oil flow and ensure a stable oil transport, the gel must be broken down. This waxy oil pipeline restart problem is especially challenging when the ambient temperature is below the pour point temperature, which indicates the lowest temperature at which the liquid oil remains pourable. When solving this problem, it is necessary to estimate the pressure to break the plug of wax-oil gel. In order to predict the breakdown pressure, it is necessary to estimate the gel strength which is measured in terms of the yield stress of the gel. Breakdown of the wax-oil gel occurs when the shear stress delivered to the gel by the applied pressure exceeds the yield stress of the gel [5]. Consequently, to provide a reliable flow assurance solution for gelled pipeline restart, it is necessary to understand the formation and breakdown phenomena of wax-oil gel.

References:

[1]   Paso, K. G., Comprehensive treatise on shut-in and restart of waxy oil pipelines. Journal of Dispersion Science and Technology 2014, 35, (8), 1060-1085.

[2]   Ekweribe, C.; Civan, F., Transient Wax Gel Formation Model for Shut-In Subsea Pipelines. Journal of Energy Resources Technology 2011, 133, (3), 033001.

[3]   Hilbert, J. In Flow Assurance: Wax Deposition & Gelling in Subsea Oil Pipelines, SPE Asia Pacific Oil and Gas Conference and Exhibition, 2010; Society of Petroleum Engineers: 2010.

[4]   Lee, H. S., Computational and Rheological Study of Wax Deposition and Gelation in Subsea Pipelines. 2008.

[5]   Venkatesan, R.; Nagarajan, N.; Paso, K.; Yi, Y.-B.; Sastry, A.; Fogler, H., The strength of paraffin gels formed under static and flow conditions. Chemical Engineering Science 2005, 60, (13), 3587-3598.

8.                  Wax deposition in gas condensate fluids must be signaled. These are not crude oils, but can have just as many problems with deposition in the well string and in transport pipelines as crude oils. The wax content in gas condensate fluids can very very high, and should be signaled to the readership of MDPI Energies.

Replay to the comment #8:  We completely agree with the reviewer on this issue. The discussion on wax deposition in gas condensate fluids has been elaborated in the revised manuscript. There are some basic differences between wax precipitation from gas condensates and wax precipitation from crudes. The pressure effect on wax precipitation from petroleum liquids increases the wax appearance temperature or the cloud-point temperature (CPT). For gas condensate systems, it may have an opposite effect as shown in Figure 3. Wax precipitation from gas condensate fluids may have some unique features [1]. The most interesting feature may relate to retrograde phenomena for the precipitated solid phase [2]. Phase behaviours of the wax containing condensate gas is extremely complex. Under different temperature and pressure, gas phase, gas-liquid, gas-solid, and gas-liquid solid multiphase appear respectively [3]. Because of pressure drop at a constant temperature, wax may form, then vaporize or become a liquid with continued pressure decrease, as wax can also precipitate before liquid condensation [2]. Therefore, if the wax precipitation is ignored, the dewpoint pressure, calculated with an equation of state (EOS), will be incorrect. For the condensate liquid, wax deposit surface roughness was found to significantly contribute to the pressure drop increases, and therefore, increase in the measured pressure drop cannot be solely attributed to deposit thickness increase [4]. When the pressure is above the dew point, the solid wax is not dominated by the condensate oil but by long plate-like coarse wax crystals. When the pressure is below the dew point, the solid wax is condensed from the gas.

The wax is soluble to the condensate oil. Then, they form a transparent colloidal, glassy block material, which has a strong binding effect on the condensate oil, the condensate gas in the wellbore does not produce condensate or wax in the production process. However, since the production back pressure and surface temperature change are in the “gas-liquid-solid” area of the phase diagram during transportation, measures should be taken to prevent condensate generation and wax formation in the ground pipeline [3]. For the commingled fluid system, the simulator predictions using the Film Mass Transfer Model (FMTM) with aging reasonably matched the deposit thickness obtained from the field data. For the condensate liquid system, both FMTM and Equilibrium Model (EM) were assessed and it was confirmed that the predictions from the two models bound the wax deposition rate calculated from the field data [4]. A summary of such discussion has been added in the revised manuscript (pages 19-20, lines 584-604).

Fig. 3. Wax in ground transportation pipeline [3]

References:

[1] Firoozabadi A. Thermodynamics of hydrocarbon reservoirs: McGraw-Hill New York; 1999.

[2] Nichita DV, Goual L, Firoozabadi A. Wax precipitation in gas condensate mixtures. SPE Production Facilities

2001;16:250-9.

[3] Wang J, Zhou F, Zhang L, Huang Y, Yao E, Zhang L, et al. Experimental study of wax deposition pattern concerning deep condensate gas in Bozi block of Tarim Oilfield and its application. Thermochimica Acta. 2019;671:1-9.

[4] Singh A, Lee H, Singh P, Sarica C. Study of the Effect of Condensate Tie-back on Wax Deposition in an Indonesian Offshore Crude Oil Pipeline.  Offshore Technology Conference: Offshore Technology Conference; 2014.

Specific line comments:

§     Line 36. Not all petroleum fluids contain wax, so line 36-37 should be modified.

Reply to specific comment #1: The sentence has been modified accordingly in the revised manuscript (page 1, lines 36-37).

§     Line 41. Oil gelatination should be gelation. Flow tendencies should be flowability

Reply to specific comment #2: The comment has been incorporated in the revised manuscript (page 1, lines 41, 42).

§     Line 42. Waxes are non-polar, not polar. Wax itself does not contain aromatic/napthenic hydrocarbons, regardless of MW.

Reply to specific comment #3: The sentence has been modified accordingly in the revised manuscript (page 1, lines 42, 43).

§     Line 64. The statement "Thus, micro-crystalline wax has a higher tendency to form stiff gel than macro-crystalline wax" is highly misleading. The micro-crystalline wax usually crystallizes at a higher temperature, but, so at high T the micro-crystalline wax may form stronger gels, but a low temperatures the macro-components usually form stronger gels at low solid fractions, due to percolation considerations and the fact that micro-components are generally lower in composition.

Reply to specific comment #4: The sentence has been modified accordingly in the revised manuscript (page 5, lines 179-181).

§     Line 67. Propylene cannot be described as having one pair of carbon atoms. Also, NM is unacceptable for an MDPI journal. Also, combustion properties are not within the scope of the manuscript title.

Reply to specific comment #5: The comment has been incorporated in the revised manuscript (Table 1).

§     Line 91-92. This is incorrect.

Reply to specific comment #6: The sentence has been deleted.

§     Line 98. This is incorrect.

Reply to specific comment #7: The sentence has been deleted.

§     Line 102. The acetone precipitation method for wax must be signaled. The spectroscopic methods are not used very often for wax content determination, and are most likely unreliable.

Reply to specific comment #8: The acetone precipitation method has been incorporated in the revised manuscript (page 8, lines 247-249).

§     Line 134. "When the temperature drops below the pour point, deposition of paraffin crystals is initiated" This statement is simply not true. The PP has no relation to the deposition onset. Please see the accepted conditions for wax deposition provided in the following book: Wax Deposition: Experimental Characterizations, Theoretical Modelling, and Field Practices, First Edition, Authors: Zhenyu Huang, Sheng Zheng, H. Scott Fogler. The conditions for wax deposition are exactly the same in the pipeline and in the reservoir, with the reservoir bulk fluid representing the bulk fluid in the pipeline.

Reply to specific comment #9: Thank you for your helpful comment. A new sub-section “1.1. Wax deposition” has been incorporated in the revised manuscript to cover the conditions of wax deposition (page 2, lines 55-83).

§     Line 137-144. It should be specifically mentioned that hot oiling dissolves paraffin components, increasing the wax content, and then subsequent cooling of the initial hot oil causes subsequent wax deposition.

Reply to specific comment #10: Thank you. The comment has been incorporated in the revised manuscript (page 11, lines 308-312).

§     Line 148. Agglutination is not a word.

Reply to specific comment #11: The word” Agglutination” has been modified in the revised manuscript (page 13, lines 354).

§     Line 180. We agree that chemical methods are difficult to standardize.

Reply to specific comment #12: Thank you.

§     Line 205. The acetone precipitation method is much more common than ether/ethanol precipitation method. The -20 degrees Celsius specification is correct.

Reply to specific comment #13: Thank you.

§     Line 247-256. Please see the major concern #3 above.

Reply to specific comment #14: We agree with the reviewer in that, the discussion on lower critical carbon number and upper critical carbon number should be signaled in the manuscript. Such information has been now cleared in the revised manuscript (pages 13-14, lines 385-397).

§     Line 262. "The WAT of micro-crystalline waxes is 4 to 6 times higher than that of macro-crystalline waxes" Generally, this statement is incorrect and should be removed. It is highly misleading.

Reply to specific comment #15: The sentence has been deleted.

§     Line 284. We like the images. Excellent!

Reply to specific comment #16: Thank you.

§     Line 296-300. We agree with this statement. This sub-field warrants additional research.

Reply to specific comment #17: Thank you.

§     Line 305. This given window for carbon and hydrogen is too narrow and thereby incorrect, and in any case must be qualified and contextualized.

Reply to specific comment #18: The sentence has been modified accordingly in the revised manuscript.

§     Line 317-318. "The amount of wax increases with increased initial crude weight" This statement is incorrect and misleading. Heavy crude oil and extra heavy crude oils often times have no wax or very little wax.

Reply to specific comment #19: The sentence has been modified accordingly in the revised manuscript (page 21, lines 609-610).

In conclusion, the authors have an incorrect understanding of wax deposition on many levels, and the manuscript is highly misleading in many ways. The authors must perform a more comprehensive literature review. A good starting point for a comprehensive literature review is the book by Zhenyu Huang listed above, which happens to be the only book entirely about wax deposition, even though some other book chapters exist that are devouted to wax deposition. Also, a more comprehensive literature review of WAT, HTGC, and anti-wax coatings is warranted. In current form, this manuscript is a disservice to the scientific community and should be completely rejected due to the abundance of misleading statements, especially concerning compositional parameters, which are known to vary widely from field to field. Our true feeling is that it would be somewhat unethical to publish this manuscript in current form, due to the abundance of misleading statements. We ask that this manuscript is completely rejected, and then subsequently re-submitted by the authors after a more comprehensive literature review is performed. We feel that it is very important to give the authors a fresh start by rejecting and resubmitting. In this way the authors can choose an overall better structure for the manuscript, and will not be burdened by having to referring back to an irrelevant version of the manuscript. For example, the authors can completely discard the unrelated sections concerning combustion, refined petroleum derivatives ethylene/propylene, and etc.

(Also this approach will give the authors more flexibility in elevating the grammatical level of the manuscript

Reply to conclusive comment: Thank you for your opinion. We have done our best to revise the manuscript to fully reflect the provided comments.

We hope that we have addressed all reviewers’ comments sufficiently. Again, we greatly appreciate you for handling our manuscript and look forward to hearing from you regarding this submission of revised manuscript.

Yours sincerely,

Byong-Hun Jeon, Ph.D.

Response to review of manuscript (energies-433124R1)

Dear Prof. Lily Peng,

We would like to thank the Editor for providing a chance to our manuscript entitled “Occurrence and characterization of paraffin wax formed in developing wells and pipelines”, for possible publication. We would like to extend our gratitude towards the reviewers for their time and effort to review our manuscript; and appreciate the reviewer’s comments that would certainly improve the quality of the manuscript. The responses to the comments are provided point by point as raised by editor and reviewers. The modifications made in the revised manuscript were highlighted in red color. We are sure that this would satisfy reviewer’s concerns.

Please find below the responses to the corrections raised by the reviewers, along with the list of changes that we have made in the revised manuscript. The permission has been obtained for the use of copyrighted materials from other sources as the author guidelines of Energies. In addition to the changes suggested by the reviewers, we have also corrected some grammatical errors in the manuscript.

Reviewer # 3

In the manuscript "Occurrence and characterization of paraffin wax formed in developing wells and pipelines", the authors El-Dalatony and co-workers review the phenomenon of wax deposition in reservoirs and tubing. The title of the manuscript does not reflect the contents of the paper, and should be broadened to also include reservoir phenomenon.

The most important change required in the manuscript is to include the conditions of wax deposition, that the depositing surface should be colder than the WAT, and there must be a negative radial thermal gradient in the pipeline, or an analogous mathematical difference in the temperature (and by inference, solubility) of the waxy reservoir oil and the reservoir rocks.

Reply to general comment: Thank you for your helpful comment. A new sub-section “1.1. Wax deposition” has been added in the revised manuscript to cover the conditions of wax deposition (page 2, lines 55-83).

Major concerns:

1.                  Each composition percentage value in the entire manuscript must be quality assured, contextualized and appropriately cited. For example, the 32.5% value of wax content listed on the first page, line 39, is incorrect, and many of the other compositional values in the manuscript are incorrect as well. We cannot approve a manuscript with incorrect compositional values. The wax content varies from field to field more than from country to country. Also, the composition of napthenatic acids, sulfur components, and etc. varies from field to field, and even varies within portions of field reservoirs, given a lack of flow communication between internal zones of specific field reservoirs. Hence, all of the compositional values given in the paper must be put into proper context and substantially improved. We understand that this is an ethical requirement for publication in MDPI Energies.

Reply to the comment #1: We agree with reviewer’s comment. Wax content may vary not only in the region but also in the same reservoir. Therefore, it is necessary to estimates it in a specific area, reservoir, or field. We have checked and corrected the compositional values in each part in the revised manuscript (lines 39 and 460).

2.                  Wax deposition will depend on the surface chemistry of the reservoir rock (if occurring in a reservoir) or the surface chemistry of the inside wall of the tubing. Metallic surfaces typically have an oxide layer at the outer surface that attracts an adsorbed water film that hinders the incipient deposition of wax on the surface. The micro-crystalline wax can more easily overcome this hindrance and deposit despite the hydrophilic resistance. The macro-crystalline types of wax are more hindered by the adsorbed water film. Similarly, if fluorinated surfaces are used, the electronegativity of the fluorine atom holds the electrons in the SP3 hybridized orbital very tight, which minimized the Debye van der Waals forces (permanent-induced) and the London van der Waals forces (induced-induced) with the paraffin wax molecules. The Keesom forces are not relevant due to the non-polar nature of the paraffin wax molecules. Hence, fluorinated surfaces minimize the attractive forces between the surface and the wax molecules, whether in soluble or crystallized form, and provide a greater resistance to incipient wax deposition than the metal oxide surfaces with the adsorbed water film. The strong implication of the surface chemistry is that fluorinated surfaces (including fluorinated coating products from DuPont, etc.) are able to in many cases completely prevent wax deposition on solid surfaces. Field experience from the Americas has shown that fluorinated coatings have been able to completely prevent wax deposition in wells. Usually the fluorinated surface will be able to prevent the deposition of wax up to a given compositional driving force, established on a solubility difference threshold. We suggest that the authors include relevant discussions about the influence of surface chemistry of the deposition surface on the incipient wax deposition. Please see the following book chapter: Encyclopedia of Surface and Colloid Science: Wax Deposition Prevention by Surface Modifications: Materials Evaluation. CRC Press. 2012. Also please see the following book chapter:  Handbook of Surface and Colloid Chemistry: Wax Deposition Investigations with Thermal Gradient Quartz Crystal Microbalance. 2008.

Replay to the comment #2:  Thanks to the reviewer for providing us this very important piece of information. Relevant discussions about the effect of surface chemistry of the reservoir rock, wall of the tubing, and composition of crude-oil have been incorporated in revised manuscript. Authors considered the recommended references by the reviewer and the available literature. Several curial factors (including surface chemistry of reservoir rock, inside wall of the tubing, and crude-oil composition) influence the incipient wax deposition. Therefore, several studies have been performed to understand the interactions among composition of reservoir rocks (such carbonate rock or/and sandstone), tubing wall materials (metallic, ceramic or micanite), composition of crude-oil (organic acids, salts, alcohols, and other natural surface-active agents), as well as types of produced wax (macro-crystalline wax and micro-crystalline wax). Such studies need be discussed for improvement of the oil recovery from the reservoir.

Two major factors have been reported to affect the wettability: 1) rocks surface morphology and 2) the intermolecular surface forces among the three phases (i.e., rock, oil and brine). Irrespective of the morphology, the wettability of the system is determined by the relative magnitude of the forces of interaction between the two liquid phases and the rock surface. The rock reservoirs were commonly categorized as oil-wet, water-wet or intermediate-wet based on the affinity of the rock surface toward oil or water phase [1]. The wettability of the system is determined by the relative magnitude of the forces of interaction between the two liquid phases and the rock surface. Such interactions and surface energies are usually classified into two classes: a) Lifshitz-van der Waals interactions (non-polar) and b) acid-base interactions (polar). Therefore, there are two indirect approaches usually used to evaluate the surface energy of solids (vapor adsorption measurements via probe vapors) and wetting (contact angle) measurements using probe liquids. Arsalan et al. [1] established a new method to quantify these fundamental interactions through characterization of the surface energetics of carbonate rocks some and sandstone by inverse gas chromatography.

A surfactant molecule has two functional groups, namely a hydrophilic (water-soluble) or polar group and a hydrophobic (oil-soluble) or non-polar group. The hydrophobic group is usually a long hydrocarbon chain (C8-C18), which may or may not be branched, while the hydrophilic group is formed by moieties such as carboxylates, sulfates, sulfonates (anionic), alcohols, polyoxyethylenated chains (nonionic) and quaternary ammonium salts (cationic) [2]. Crude-oil contains organic acids and salts, alcohols and other natural surface-active agents. When crude oil is brought in contact with brine or water, these natural surfactants accumulate at the interface and form an adsorbed film which lowers the interfacial tension of the crude oil/water interface. Depending on the type of crude oil, the adsorbed film at the interface can be either fluid or very viscoelastic and able to form a skin [3]. Therefore, the molecular packing, surface viscosity, surface elasticity, and surface charge of the adsorbed film are very vital parameters that determine various phenomena such as coalescence of emulsion droplets, as well as oil drop migration in porous media. It has been reported that, ceramic or micanite in-line ring heaters are recommended to be applied for prevention of wax accumulation in Tubing (Fig. 1). Apart of this discussion has been added in the revised manuscript (page 2, lines 69-83) and (page 15, lines 445-457).

Fig 1. Suggest materials to be used for creating the required temperature in the very wax accumulation zone: Ring heaters are ideal for heating tubes and cylinders (a) and energy-conserving ring clamp heater (b) [4].

References:

[1]   Arsalan, N.; Buiting, J. J.; Nguyen, Q. P. J. C.; Physicochemical, S. A.; Aspects, E., Surface energy and wetting behavior of reservoir rocks. 2015, 467, 107-112.

[2]   Paso, K.; Viitala, T.; Aske, N.; Sjöblom, J., Wax deposition prevention by surface modifications: materials evaluation. Encyclopedia of Surface and Colloid Science (second edition), CRC Press, Boca Raton 2012.

[3]   Kanicky, J. R.; Lopez-Montilla, J.-C.; Pandey, S.; Shah, D. O., Surface chemistry in the petroleum industry. Handbook of applied surface and colloid chemistry 2001, 1, 251-267.

[4]   Musipov, H.; Akhmadulin, R.; Bakanovskaya, L. In Wax Accumulation Prevention Method in Tubing, IOP Conference Series: Materials Science and Engineering, 2017; IOP Publishing: 2017; p 012018.

3.                  The two critical carbon numbers (CCN) should be signaled. The "lower critical carbon number" and the "upper critical carbon number" establish on a compositional basis the range of components that contribute to the deposition process, respectively. If the deposition surface temperature is lower than the bulk fluid temperature and the deposition surface temperature is less than the WAT, then the "lower critical carbon number" will establish a lower bound to the range of components that contribute to the deposition process. If the bulk fluid temperature is below the WAT and the deposition surface temperature is below the WAT, then the "upper critical carbon number" establishes an upper limit to the range of components that contribute to the deposition process. There is substantial literature available on the compositional basis of wax deposition.

Replay to the comment #3:  We agree with the reviewer in that, the discussion on lower critical carbon number and upper critical carbon number along with deposition surface temperature should be signaled in the manuscript. Such information has been now cleared in the revised manuscript (pages 13-14, lines 385-397). Wax deposition is effected by temperature and critical carbon number (CCN) (Fig. 2). The CCN has been reported to be C27-C28. Wax deposition occur when the deposition surface temperature is below than both the bulk fluid temperature and the wax appearance temperature (WAT).  The lower carbon number (<27) start to deposit on the pipeline surface when deposition surface temperature is lower than bulk fluid (i.e., crude-oil temperature) and WAT, resulting in micro-crystallization. Future decrease in surface deposition temperature cause macro-crystallization of upper carbon number (>28) replacing micro-crystallization. Wax molecules with upper carbon number than CCN diffused into the deposit, while those with carbon number less than CCN diffused out from the deposit and increases the hardness of deposit (this process is called “aging” or “nucleation”) [1]. Initially, the rate of wax deposition is high, but it slows down as more wax is deposited on the pipe surface. The thickness of the wax layer (macro- crystallization) increases and acts as thermal insulation factor. The wax deposition and its thickness control by several mechanism such as Shear dispersion, Brownian diffusion, gravity settling, and molecular diffusion [2].

Fig. 2. Changes in carbon number distribution of deposit with time (Toil=WAT+ 5 °C, TCoolant= WAT -10 °C) [3].

References

[1] Huang, Z.; Lee, H. S.; Senra, M.; Scott Fogler, H. J. A. J., A fundamental model of wax deposition in subsea oil pipelines. 2011, 57, (11), 2955-2964.

[2] Roehner, R.; Fletcher, J.; Hanson, F.; Dahdah, N. J. E.; fuels, Comparative compositional study of crude oil solids from the Trans Alaska Pipeline System using high-temperature gas chromatography. 2002, 16, (1), 211-217.

[3] Quan, Q.; Gong, J.; Wang, W.; Gao, G., Study on the aging and critical carbon number of wax deposition with temperature for crude oils. Journal of Petroleum Science and Engineering 2015, 130, 1-5.

4.      Mathematical modelling of wax deposition using convective heat and mass transfer correlations must be signaled, considering that the scope of the title includes the occurrence of paraffin wax formation. There exists an abundance of literature available on this topic using a simple google search.

Replay to the comment #4:  Thank you for your helpful comment. We have added a new sub-section entitled “1.2.    Mathematical modelling of wax deposition” in the revised manuscript (pages 3-4, lines 85-163). This sub-section includes the following information:

There are four wax deposition mechanisms need to be taken into account for modeling the wax deposit growth in pipelines, i.e. molecular diffusion, shear dispersion, gravity settlement and Brownian diffusion[1, 2]. The molecular diffusion, which can be described by taking the mass transfer and energy balance into account, is considered as the dominant mechanism during the wax deposition process [1, 3-6]. Burger, Perkins and Striegler [1] and Weingarten and Euchner [7] proposed that the shear dispersion mechanism needs to be incorporated for the wax deposition behavior analysis, especially in laminar flow. However, Brown, Niesen and Erickson [4] performed a set of experiments and pointed out that the mechanism does not take a role during the wax deposition. The gravity settling mechanism suggests that the wax crystals are would be precipitated to the bottom of the pipelines as the crystals are denser than the oil. However, it appears that the effect is negligible as experimental studies comparing vertical and horizontal fluid flows did not reveal any difference in the deposited wax amount [8]. As soon as the wax crystals are precipitated and suspended in the oil, the wax crystals will behave with Brownian motion. Since the motion effect is likely to transport the crystals to the area with lower wax concentration, it may have an influence on the wax deposition behavior. However, its effect is generally considered minimal and it is generally neglected during modeling the overall deposition mechanisms [8]. Consequently, the latter three mechanisms are widely not accepted, especially for the computational modeling procedures [8-14]. In this part of the study, the most dominant mechanism, the molecular diffusion, is briefly reviewed.

In order to incorporate the molecular dispersion mechanisms, the radial temperature gradient of the pipeline needs to be taken into account. This is because a radial convective flux of wax molecules is induced by the concentration gradients, which strongly depends on the temperature gradient inside the pipeline. Therefore, the flux can be calculated by difference in the wax concentration between the bulk and the interface of the oil and the wax deposits. The mass transfer can be expressed with the equation as follows,

where,  is the clean pipe radius,  is the flowable radius,  is weight fraction of solid in the wax deposit,  is the pipeline length,  is the wax deposit density,  is the convective mass transfer coefficient,  and  are the dissolved paraffin concentration in the bulk and at the oil-gel interface, respectively.

Another process that needs to be incorporated is the growth of the deposited wax. Since there exists a temperature gradient within the deposit, an internal mass transfer also occurs. The wax deposition growth can be described by the equation as follows,

where,  is effective diffusivity of wax in the waxy given by Cussler, et al. [15]:

where,  is the molecular diffusivity of wax in oil and  is the aspect ratio of the wax crystals in the deposit.

Within the system the energy balance is met as the heat conduction across the deposit thickness is the sum of the radial convective heat flux and the latent heat of solidification. Therefore, the energy balance can be expressed with the equation as follows,

where, is the temperature of oil at the center,  is the oil-wax interface temperature,  is the ambient temperature, is the conductive heat coefficient,  is the inner convective heat transfer coefficient and  is the heat of wax solidification.

Assuming flow in the pipe is laminar, the heat transfer coefficients can be determined by the empirical correlations, such as the Hausen correlation [16] and the Seider and Tate correlation [17]. The former is valid for long tubes, while the latter shows a good agreement with a short tube length [6]. The Nusselt number for the Graetz number lower than 100 can be calculated by the Hausen correlation as follows,

where,  and  are Graetz number for heat transfer and for mass transfer, respectively. When the Graetz number is larger than 100, the Nusselt number can be obtained by the Seider and Tate correlation as follow,

The heat transfer coefficient and the mass transfer coefficient can be then calculated by the equations as follows,

where,  is the thermal conductivity of the oil.

The computational modeling can be a powerful tool to predict and to interpret the wax deposition phenomena after validated with available experiments data. However, most of the lab-scale experiments are based on the single-phase flow, which frequently yields deviated outcomes when compared to more-general multiphase flow cases. In addition, Leporini et al. [17] emphasized that the modeling results are not always applicable for the field scale analysis and that the scale effect also needs to be taken into account for more reliable interpretation. Consequently, recent studies focus on more complex environments, such as wax deposition behavior under multi-phase flow and turbulent flow.

References:

[1]         Burger, E. D.; Perkins, T. K.; Striegler, J. H., Studies of Wax Deposition in the Trans Alaska Pipeline. SPE-8788-PA 1981, 33, (06), 1075-1086.

[2]         Todi, S. Experimental and modeling studies of wax deposition in crude oil carrying pipelines. University of Utah, Utah, 2005.

[3]         Bern, P. A.; Withers, V. R.; Cairns, R. J. R., Wax Deposition in Crude Oil Pipelines. In European Offshore Technology Conference and Exhibition, Society of Petroleum Engineers: London, United Kingdom, 1980; p 8.

[4]         Brown, T. S.; Niesen, V. G.; Erickson, D. D., Measurement and Prediction of the Kinetics of Paraffin Deposition. In SPE Annual Technical Conference and Exhibition, Society of Petroleum Engineers: Houston, Texas, 1993; p 16.

[5]         Venkatesan, R.; Fogler, H. S., Comments on analogies for correlated heat and mass transfer in turbulent flow. AIChE Journal 2004, 50, (7), 1623-1626.

[6]         Singh, P.; Venkatesan, R.; Fogler, H. S.; Nagarajan, N., Formation and aging of incipient thin film wax-oil gels. AIChE Journal 2000, 46, (5), 1059-1074.

[7]         Weingarten, J. S.; Euchner, J. A., Methods for Predicting Wax Precipitation and Deposition. SPE-15654-PA 1988, 3, (01), 121-126.

[8]         Guo, B.; Song, S.; Ghalambor, A.; Lin, T. R., Chapter 15 - Flow Assurance. In Offshore Pipelines (Second Edition), Guo, B.; Song, S.; Ghalambor, A.; Lin, T. R., Eds. Gulf Professional Publishing: Boston, 2014; pp 179-231.

[9]         Lee, H. S. COMPUTATIONAL AND RHEOLOGICAL STUDY OF WAX DEPOSITION AND GELATION IN SUBSEA PIPELINES. University of Michigan, 2008.

[10]     Zheng, S.; Saidoun, M.; Mateen, K.; Palermo, T.; Ren, Y.; Fogler, H. S., Wax Deposition Modeling with Considerations of Non-Newtonian Fluid Characteristics. In Offshore Technology Conference, Offshore Technology Conference: Houston, Texas, USA, 2016; p 18.

[11]     Leporini, M.; Terenzi, A.; Marchetti, B.; Giacchetta, G.; Corvaro, F., Experiences in numerical simulation of wax deposition in oil and multiphase pipelines: Theory versus reality. Journal of Petroleum Science and Engineering 2019, 174, 997-1008.

[12]     Giacchetta, G.; Marchetti, B.; Leporini, M.; Terenzi, A.; Dall’Acqua, D.; Capece, L.; Cocci Grifoni, R., Pipeline wax deposition modeling: A sensitivity study on two commercial software. Petroleum 2017.

[13]     Li, S.; Huang, Q.; Zhao, D.; Lv, Z., Relation of heat and mass transfer in wax diffusion in an emulsion of water and waxy crude oil under static condition. Experimental Thermal and Fluid Science 2018, 99, 1-12.

[14]     Soedarmo, A. A.; Daraboina, N.; Sarica, C., Validation of wax deposition models with recent laboratory scale flow loop experimental data. Journal of Petroleum Science and Engineering 2017, 149, 351-366.

[15]     Cussler, E. L.; Hughes, S. E.; Ward, W. J.; Aris, R., Barrier membranes. Journal of Membrane Science 1988, 38, (2), 161-174.

[16]     Hausen, H., Darstellung des warmeuberganges in rohren durch verallgemeinerte potenz-beziehungen. ZVDI Beih Verfahrenstech 1943, 4, 91.

[17]     Sieder, E. N.; Tate, G. E., Heat Transfer and Pressure Drop of Liquids in Tubes. Industrial & Engineering Chemistry 1936, 28, (12), 1429-1435.

5.                  HTGC methods for characterization of the wax should be signaled. These methods are very commonly used in industry. It is very important to signal HTGC because this is the most common method and easiest method to characterize the wax itself. Most paraffinic crude oils can be injected directly into the HTGC without prior separation of the asphaltenes, because the asphaltenes do not have a vapor pressure. However, even after separation of the asphaltenes, the HTGC method can still be used. The FID is the most common detector for HTGC. New 2D-HTGC techniques may also be useful for understanding the structure of the wax molecules. Other methods for establishing the number of rings on the wax structure can also be signaled in the manuscript.

Replay to the comment #5:  We agree with reviewer’s comments that high temperature gas chromatography (HTGC) is one of the most effective method for wax determination. The distinction between general GC and HTGC is not clearly defined, but GC at 340 degrees or more can be called HTGC. HTGC enables the analysis of a wide range of molecular weights from C7 to > C100 [1]. Recently, two-dimensional GC with flame ionization detection (HT-GCxGC-FID) or a time-of-flight mass spectrometer (HT-GCxGC-TOFMS) has been used to separate n-paraffin, iso-paraffin, and cyclo-praffin [2]. Such information has been incorporated in the revised manuscript (page 15, lines 461-467).

References:

[1]   França, D.; Pereira, V. B.; Coutinho, D. M.; Ainstein, L. M.; Azevedo, D. A., Speciation and quantification of high molecular weight paraffins in Brazilian whole crude oils using high-temperature comprehensive two-dimensional gas chromatography. Fuel 2018, 234, 1154-1164.

[2]   Mahé, L.; Courtiade, M.; Dartiguelongue, C.; Ponthus, J.; Souchon, V.; Thiébaut, D., Overcoming the high-temperature two-dimensional gas chromatography limits to elute heavy compounds. Journal of Chromatography A. 2012, 1229, 298-301.

6.                  Because the WAT determines where the wax will deposit, measurement methods for the WAT must be signaled in the manuscript. This is very important for a practical perspective for the readership of MDPI Energies journal.

Replay to the comment #6:  Thank you for your valuable comment. As recommended by reviewer, brief explanations of industry established techniques for measuring WAT has been incorporated in the revised manuscript as the following:

WAT is the temperature at which wax crystal formation begins. The formation of wax causes the fluid to become cloudy, hence WAT is also called cloud point. The industry established techniques for measuring WAT in the laboratory are ranging from ASTM D2500, cross polar microscopy (CPM), light transmission (LT), differential scanning calorimetry (DSC), viscometry, and filter plugging infrared (FTIR) to cold finger test [1, 2]. The resultant values of WAT depend on the measurement techniques. The ASTM method relies on the visual observation of wax crystals, which reduces the reliability of WAT measurements of opaque or dark oils. Therefore, CPM, LT, DSC, or viscometry methods are preferred for WAT measurements of dark oils. DSC and viscometry are affected by the amount of precipitated wax, while light transmittance (LT) and light scattering are affected by the number of wax crystals. WAT measurement methods have advantages and disadvantages, so it is recommended to use multiple technologies together depending on the given conditions and circumstances [3, 4]. Such explanation has been incorporated in the revised manuscript (page 7, lines 197-208).

References:

[1]   Uba, E.; Ikeji, K.; Onyekonwu, M. In Measurement of wax appearance temperature of an offshore live crude oil using laboratory light transmission method, Nigeria Annual International Conference and Exhibition, 2004; Society of Petroleum Engineers: 2004.

[2]   Mmata, B.; Onyekonwu, M. In Measurement of the Wax Appearance Temperature of a Gas Condensate Using High Pressure Microscopy Technique, SPE Nigeria Annual International Conference and Exhibition, 2018; Society of Petroleum Engineers: 2018.

[3]   Monger-McClure, T.; Tackett, J.; Merrill, L., Comparisons of cloud point measurement and paraffin prediction methods. SPE production & facilities 1999, 14, (01), 4-16.

7.                  Paraffin wax can cause problems with wax deposition as well as with plugging of pipelines after flow shut-in for planned maintenance or emergency contingency situations. Hence, this related problem of wax gelling in the pipeline after shut-in and pipeline restart after gelling should be briefly mentioned in the manuscript, because several of the analysis techniques and scientific/engineering considerations are identical for both related industrial problems. In fact, it can be shown by TG-QCM that the incipient deposition mechanism is actually gelation. (Please see the book chapter above on the Thermal Gradient Quartz Crystal Microbalance).

Replay to the comment #8:  Thank you for your helpful comment. The related problem of wax gelling in the pipeline after gelling has been discussed in the manuscript (page 14, lines 398-411). We have added a new sub-section entitled 2.3. Effects of wax deposition on the flow assurance of hydrocarbon” in the revised manuscript to deal with flow assurance problems in detail. The pipeline restart problem by wax gelling has been incorporated in the last paragraph of the sub-section as the following:

Wax precipitation lead to wax deposition and flow restriction during the oil flow, which consequently cause wax precipitation during production also causes problems when trying to restart the flow [1-5]. If oil production is halted by a scheduled maintenance or an emergency situation, the temperature and solubility of wax decreases in a static condition. In this circumstances, wax-oil gel is formed due to precipitation of wax molecules, which can lead to blockage of the entire pipeline. In order to restart the oil flow and ensure a stable oil transport, the gel must be broken down. This waxy oil pipeline restart problem is especially challenging when the ambient temperature is below the pour point temperature, which indicates the lowest temperature at which the liquid oil remains pourable. When solving this problem, it is necessary to estimate the pressure to break the plug of wax-oil gel. In order to predict the breakdown pressure, it is necessary to estimate the gel strength which is measured in terms of the yield stress of the gel. Breakdown of the wax-oil gel occurs when the shear stress delivered to the gel by the applied pressure exceeds the yield stress of the gel [5]. Consequently, to provide a reliable flow assurance solution for gelled pipeline restart, it is necessary to understand the formation and breakdown phenomena of wax-oil gel.

References:

[1]   Paso, K. G., Comprehensive treatise on shut-in and restart of waxy oil pipelines. Journal of Dispersion Science and Technology 2014, 35, (8), 1060-1085.

[2]   Ekweribe, C.; Civan, F., Transient Wax Gel Formation Model for Shut-In Subsea Pipelines. Journal of Energy Resources Technology 2011, 133, (3), 033001.

[3]   Hilbert, J. In Flow Assurance: Wax Deposition & Gelling in Subsea Oil Pipelines, SPE Asia Pacific Oil and Gas Conference and Exhibition, 2010; Society of Petroleum Engineers: 2010.

[4]   Lee, H. S., Computational and Rheological Study of Wax Deposition and Gelation in Subsea Pipelines. 2008.

[5]   Venkatesan, R.; Nagarajan, N.; Paso, K.; Yi, Y.-B.; Sastry, A.; Fogler, H., The strength of paraffin gels formed under static and flow conditions. Chemical Engineering Science 2005, 60, (13), 3587-3598.

8.                  Wax deposition in gas condensate fluids must be signaled. These are not crude oils, but can have just as many problems with deposition in the well string and in transport pipelines as crude oils. The wax content in gas condensate fluids can very very high, and should be signaled to the readership of MDPI Energies.

Replay to the comment #8:  We completely agree with the reviewer on this issue. The discussion on wax deposition in gas condensate fluids has been elaborated in the revised manuscript. There are some basic differences between wax precipitation from gas condensates and wax precipitation from crudes. The pressure effect on wax precipitation from petroleum liquids increases the wax appearance temperature or the cloud-point temperature (CPT). For gas condensate systems, it may have an opposite effect as shown in Figure 3. Wax precipitation from gas condensate fluids may have some unique features [1]. The most interesting feature may relate to retrograde phenomena for the precipitated solid phase [2]. Phase behaviours of the wax containing condensate gas is extremely complex. Under different temperature and pressure, gas phase, gas-liquid, gas-solid, and gas-liquid solid multiphase appear respectively [3]. Because of pressure drop at a constant temperature, wax may form, then vaporize or become a liquid with continued pressure decrease, as wax can also precipitate before liquid condensation [2]. Therefore, if the wax precipitation is ignored, the dewpoint pressure, calculated with an equation of state (EOS), will be incorrect. For the condensate liquid, wax deposit surface roughness was found to significantly contribute to the pressure drop increases, and therefore, increase in the measured pressure drop cannot be solely attributed to deposit thickness increase [4]. When the pressure is above the dew point, the solid wax is not dominated by the condensate oil but by long plate-like coarse wax crystals. When the pressure is below the dew point, the solid wax is condensed from the gas.

The wax is soluble to the condensate oil. Then, they form a transparent colloidal, glassy block material, which has a strong binding effect on the condensate oil, the condensate gas in the wellbore does not produce condensate or wax in the production process. However, since the production back pressure and surface temperature change are in the “gas-liquid-solid” area of the phase diagram during transportation, measures should be taken to prevent condensate generation and wax formation in the ground pipeline [3]. For the commingled fluid system, the simulator predictions using the Film Mass Transfer Model (FMTM) with aging reasonably matched the deposit thickness obtained from the field data. For the condensate liquid system, both FMTM and Equilibrium Model (EM) were assessed and it was confirmed that the predictions from the two models bound the wax deposition rate calculated from the field data [4]. A summary of such discussion has been added in the revised manuscript (pages 19-20, lines 584-604).

Fig. 3. Wax in ground transportation pipeline [3]

References:

[1] Firoozabadi A. Thermodynamics of hydrocarbon reservoirs: McGraw-Hill New York; 1999.

[2] Nichita DV, Goual L, Firoozabadi A. Wax precipitation in gas condensate mixtures. SPE Production Facilities

2001;16:250-9.

[3] Wang J, Zhou F, Zhang L, Huang Y, Yao E, Zhang L, et al. Experimental study of wax deposition pattern concerning deep condensate gas in Bozi block of Tarim Oilfield and its application. Thermochimica Acta. 2019;671:1-9.

[4] Singh A, Lee H, Singh P, Sarica C. Study of the Effect of Condensate Tie-back on Wax Deposition in an Indonesian Offshore Crude Oil Pipeline.  Offshore Technology Conference: Offshore Technology Conference; 2014.

Specific line comments:

§     Line 36. Not all petroleum fluids contain wax, so line 36-37 should be modified.

Reply to specific comment #1: The sentence has been modified accordingly in the revised manuscript (page 1, lines 36-37).

§     Line 41. Oil gelatination should be gelation. Flow tendencies should be flowability

Reply to specific comment #2: The comment has been incorporated in the revised manuscript (page 1, lines 41, 42).

§     Line 42. Waxes are non-polar, not polar. Wax itself does not contain aromatic/napthenic hydrocarbons, regardless of MW.

Reply to specific comment #3: The sentence has been modified accordingly in the revised manuscript (page 1, lines 42, 43).

§     Line 64. The statement "Thus, micro-crystalline wax has a higher tendency to form stiff gel than macro-crystalline wax" is highly misleading. The micro-crystalline wax usually crystallizes at a higher temperature, but, so at high T the micro-crystalline wax may form stronger gels, but a low temperatures the macro-components usually form stronger gels at low solid fractions, due to percolation considerations and the fact that micro-components are generally lower in composition.

Reply to specific comment #4: The sentence has been modified accordingly in the revised manuscript (page 5, lines 179-181).

§     Line 67. Propylene cannot be described as having one pair of carbon atoms. Also, NM is unacceptable for an MDPI journal. Also, combustion properties are not within the scope of the manuscript title.

Reply to specific comment #5: The comment has been incorporated in the revised manuscript (Table 1).

§     Line 91-92. This is incorrect.

Reply to specific comment #6: The sentence has been deleted.

§     Line 98. This is incorrect.

Reply to specific comment #7: The sentence has been deleted.

§     Line 102. The acetone precipitation method for wax must be signaled. The spectroscopic methods are not used very often for wax content determination, and are most likely unreliable.

Reply to specific comment #8: The acetone precipitation method has been incorporated in the revised manuscript (page 8, lines 247-249).

§     Line 134. "When the temperature drops below the pour point, deposition of paraffin crystals is initiated" This statement is simply not true. The PP has no relation to the deposition onset. Please see the accepted conditions for wax deposition provided in the following book: Wax Deposition: Experimental Characterizations, Theoretical Modelling, and Field Practices, First Edition, Authors: Zhenyu Huang, Sheng Zheng, H. Scott Fogler. The conditions for wax deposition are exactly the same in the pipeline and in the reservoir, with the reservoir bulk fluid representing the bulk fluid in the pipeline.

Reply to specific comment #9: Thank you for your helpful comment. A new sub-section “1.1. Wax deposition” has been incorporated in the revised manuscript to cover the conditions of wax deposition (page 2, lines 55-83).

§     Line 137-144. It should be specifically mentioned that hot oiling dissolves paraffin components, increasing the wax content, and then subsequent cooling of the initial hot oil causes subsequent wax deposition.

Reply to specific comment #10: Thank you. The comment has been incorporated in the revised manuscript (page 11, lines 308-312).

§     Line 148. Agglutination is not a word.

Reply to specific comment #11: The word” Agglutination” has been modified in the revised manuscript (page 13, lines 354).

§     Line 180. We agree that chemical methods are difficult to standardize.

Reply to specific comment #12: Thank you.

§     Line 205. The acetone precipitation method is much more common than ether/ethanol precipitation method. The -20 degrees Celsius specification is correct.

Reply to specific comment #13: Thank you.

§     Line 247-256. Please see the major concern #3 above.

Reply to specific comment #14: We agree with the reviewer in that, the discussion on lower critical carbon number and upper critical carbon number should be signaled in the manuscript. Such information has been now cleared in the revised manuscript (pages 13-14, lines 385-397).

§     Line 262. "The WAT of micro-crystalline waxes is 4 to 6 times higher than that of macro-crystalline waxes" Generally, this statement is incorrect and should be removed. It is highly misleading.

Reply to specific comment #15: The sentence has been deleted.

§     Line 284. We like the images. Excellent!

Reply to specific comment #16: Thank you.

§     Line 296-300. We agree with this statement. This sub-field warrants additional research.

Reply to specific comment #17: Thank you.

§     Line 305. This given window for carbon and hydrogen is too narrow and thereby incorrect, and in any case must be qualified and contextualized.

Reply to specific comment #18: The sentence has been modified accordingly in the revised manuscript.

§     Line 317-318. "The amount of wax increases with increased initial crude weight" This statement is incorrect and misleading. Heavy crude oil and extra heavy crude oils often times have no wax or very little wax.

Reply to specific comment #19: The sentence has been modified accordingly in the revised manuscript (page 21, lines 609-610).

In conclusion, the authors have an incorrect understanding of wax deposition on many levels, and the manuscript is highly misleading in many ways. The authors must perform a more comprehensive literature review. A good starting point for a comprehensive literature review is the book by Zhenyu Huang listed above, which happens to be the only book entirely about wax deposition, even though some other book chapters exist that are devouted to wax deposition. Also, a more comprehensive literature review of WAT, HTGC, and anti-wax coatings is warranted. In current form, this manuscript is a disservice to the scientific community and should be completely rejected due to the abundance of misleading statements, especially concerning compositional parameters, which are known to vary widely from field to field. Our true feeling is that it would be somewhat unethical to publish this manuscript in current form, due to the abundance of misleading statements. We ask that this manuscript is completely rejected, and then subsequently re-submitted by the authors after a more comprehensive literature review is performed. We feel that it is very important to give the authors a fresh start by rejecting and resubmitting. In this way the authors can choose an overall better structure for the manuscript, and will not be burdened by having to referring back to an irrelevant version of the manuscript. For example, the authors can completely discard the unrelated sections concerning combustion, refined petroleum derivatives ethylene/propylene, and etc.

(Also this approach will give the authors more flexibility in elevating the grammatical level of the manuscript

Reply to conclusive comment: Thank you for your opinion. We have done our best to revise the manuscript to fully reflect the provided comments.

We hope that we have addressed all reviewers’ comments sufficiently. Again, we greatly appreciate you for handling our manuscript and look forward to hearing from you regarding this submission of revised manuscript.

Yours sincerely,

Byong-Hun Jeon, Ph.D.

Round 2

Reviewer 3 Report

We highly appreciate the spirit of the multitude of changes made to the manuscript, and we also appreciate the exhaustive responses given to our initial comment.

A few more important changes are required to the manuscript prior to publication:

The authors have not properly understood the distinction between "upper critical carbon number" and "lower critical carbon number". The "lower critical carbon number" is the classic CCN. The "upper critical carbon number" is a completely different parameter from the classic CCN, and only applies to cold flow slurry conditions, and provides an upper bound to the range of components that contribute to the wax deposition process.

We refer the authors to the following reference for the "upper CCN":

Paso, K. G., & Fogler, H. S. (2004). Bulk stabilization in wax deposition systems. Energy & fuels, 18(4), 1005-1013.

Please revise the manuscript lines 385-393 in light of a correct understanding of the "lower CCN" and the "upper CCN"

The paragraph in Line 69-83 should be expanded to include a discussion of the effect of surface wettability on wax deposition.

We refer the authors to the following reference for the surface wettability influence on wax deposition:

Paso, K., Braathen, B., Viitala, T., Aske, N., Rønningsen, H. P., & Sjöblom, J. (2008). Wax deposition investigations with thermal gradient quartz crystal microbalance. In Handbook of surface and colloid chemistry (pp. 571-588). CRC Press.

The paragraph on line 445-457 must be expanded to include a direct connection to the wax deposition problem. We direct the authors to the following book in order to find a direct connection between surfactants and dispersant additives to remediate wax deposition:

Kelland, M. A. (2014). Production chemicals for the oil and gas industry. CRC press.

With these additional changes, then the manuscript can be considered for publication.

We congradulate the authors on substantially improving the manuscript, and we wish the authors the best of luck.

Author Response

February 23rd, 2019

Editor: Energies

Response to review of manuscript (energies-433124R2)

Dear Prof. Lily Peng,

We would like to thank the Editor for providing a chance to our manuscript entitled “Occurrence and characterization of paraffin wax formed in developing wells and pipelines”, for possible publication. We would like to extend our gratitude towards the reviewer for his/her time and effort to review our revised manuscript; and appreciate the reviewer comments that would certainly improve the quality of the manuscript. The responses to the comments are provided point by point as raised by the reviewer. The modifications made in the revised manuscript were highlighted in red color. We are sure that this would satisfy reviewer concerns.

Please find below the responses to the corrections raised by the reviewer, along with the list of changes that we have made in the revised manuscript. The permission has been obtained for the use of copyrighted materials from other sources as the author guidelines of Energies. In addition to the changes suggested by the reviewers, we have also corrected some grammatical errors in the manuscript.

Reviewer # 3

We highly appreciate the spirit of the multitude of changes made to the manuscript, and we also appreciate the exhaustive responses given to our initial comment. A few more important changes are required to the manuscript prior to publication:

Reply to general comment: Thank you so for your overall evaluation and suggestions for the manuscript. We have done our best to revise the manuscript to fully reflect the reviewer comments.

1.      The authors have not properly understood the distinction between "upper critical carbon number" and "lower critical carbon number". The "lower critical carbon number" is the classic CCN. The "upper critical carbon number" is a completely different parameter from the classic CCN, and only applies to cold flow slurry conditions, and provides an upper bound to the range of components that contribute to the wax deposition process. We refer the authors to the following reference for the "upper CCN":

Paso, K. G., & Fogler, H. S. (2004). Bulk stabilization in wax deposition systems. Energy & fuels, 18(4), 1005-1013.

Please revise the manuscript lines 385-393 in light of a correct understanding of the "lower CCN" and the "upper CCN"

Replay to the comment #1: Thank you for your comment. The comment has incorporated in the revised manuscript (pages 13-14, lines 384-398).

2.      The paragraph in Line 69-83 should be expanded to include a discussion of the effect of surface wettability on wax deposition. We refer the authors to the following reference for the surface wettability influence on wax deposition:

Paso, K., Braathen, B., Viitala, T., Aske, N., Rønningsen, H. P., & Sjöblom, J. (2008). Wax deposition investigations with thermal gradient quartz crystal microbalance. In Handbook of surface and colloid chemistry (pp. 571-588). CRC Press.

Replay to the comment #3: Thank you for your suggestion. The recommended references have been added to the revised manuscript (page 2, lines 73-80).

3.      The paragraph on line 445-457 must be expanded to include a direct connection to the wax deposition problem. We direct the authors to the following book in order to find a direct connection between surfactants and dispersant additives to remediate wax deposition:

Kelland, M. A. (2014). Production chemicals for the oil and gas industry. CRC press.

Replay to the comment #4: Thank you for valuable comment. The paragraph has been expanded in the revised manuscript (page 15, lines 446-457), as well as the suggested reference has been added.

With these additional changes, then the manuscript can be considered for publication. We congratulate the authors on substantially improving the manuscript, and we wish the authors the best of luck.

Reply to conclusive comment: We would like to show our gratitude towards the reviewer for his/her time and effort to review our manuscript; and appreciate the reviewer comments that had certainly improved the quality of the manuscript.

We hope that we have addressed all reviewers’ comments sufficiently. Again, we greatly appreciate you for handling our manuscript and look forward to hearing from you regarding this submission of revised manuscript.

Yours sincerely,

Byong-Hun Jeon, Ph.D.
